# BACKGROUND PROMPT FOR FEW-SHOT OUT-OF-DISTRIBUTION DETECTION

## ABSTRACT

Existing foreground-background (FG-BG) decomposition methods for the few-shot out-of-distribution (FS-OOD) detection often suffer from low robustness due to over-reliance on the local class similarity and a fixed background patch extraction strategy. To address these challenges, we propose a new FG-BG decomposition framework, namely **Mambo**, for FS-OOD detection. Specifically, we propose to first learn a background prompt to obtain the local background similarity containing both the background and image semantic information, and then refine the local background similarity using the local class similarity. As a result, we use both the refined local background similarity and the local class similarity to conduct background extraction, reducing the dependence of the local class similarity in previous methods. Furthermore, we propose the patch self-calibrated tuning to consider the sample diversity to flexibly select numbers of background patches for different samples, and thus exploring the issue of fixed background extraction strategies in previous methods. Extensive experiments on real-world datasets demonstrate that our proposed Mambo achieves the best performance, compared to SOTA methods in terms of OOD detection and near OOD detection setting. The source code will be released at https://anonymous.4open.science/r/Mambo-CBC1.

## 1 INTRODUCTION

Out-of-distribution (OOD) detection is designed to simultaneously detect OOD samples (*i.e.*, images) and conduct downstream tasks with in-distribution (ID) samples (*i.e.*, training images), and has been widely used to produce trustworthy machine learning (Jaeger et al., 2023; Cheng et al., 2025). Considering that the advanced development of vision-language models (VLMs) (*e.g.*, CLIP (Radford et al., 2021)) and the limitation of labeled samples in real applications, few-shot OOD (FS-OOD) detection utilizes the pre-trained knowledge in VLMs to explore the issue of limitedly labeled data in ID samples, and thus effectively distinguish OOD samples from ID samples as well as conduct downstream tasks with limited ID data. Recently, FS-OOD has been becoming increasingly important in real-world scenarios (Bai et al., 2024; Jiang et al., 2024).

Previous CLIP-based FS-OOD detection methods can be classified into two categories, *i.e.*, negative prompt methods and foreground-background (FG-BG) decomposition methods. Negative prompt methods learn prompts for OOD features of all samples (*i.e.*, images) from ID samples to enlarge the similarity gap between ID and OOD samples, which helps to detect OOD samples from ID samples. For instance, ID-like (Bai et al., 2024) downsamples ID samples and uses low-similarity features as OOD features to train negative prompt. However, these methods heavily relies on the quality of the learned negative prompt, which often requires increased computational resources for optimization. In contrast, FG-BG decomposition methods aim to extract background information from local image features unrelated to the classes of all ID samples, ID classes for short. As a result, they do not incur extra computational cost, as well as reduce the interference of the background information on the model prediction. For instance, LoCoOp (Miyai et al., 2023a) removes ID-irrelevant background information from local image features to avoid the influence of background information.

Although previous FG-BG decomposition methods have achieved some progress, several limitations remain unresolved. First, these methods heavily rely on local class similarity (*i.e.*, the cosine similarity between local image features and class text features). If the local class similarity is not good enough

Figure 1: The flowchart of our proposed **Mambo**. Given an in-distribution (ID) image, we use the frozen image encoder of CLIP to obtain global and local image features. We also use the frozen text encoder of CLIP to learn the class prompt to output class text features. The cosine similarity between global image features and class text features is used to conduct downstream tasks, *e.g.*, classification. After that, we investigate a new background extraction module to extract the background. Specifically, we first learn a background prompt with the frozen text encoder of CLIP to output the background text feature, which is used to obtain the local background similarity by calculating the cosine similarity between it and the local image features. We then design the **local similarity refinement** to obtain a new local background similarity with the help of the ground truth class in the local class similarity. Next, we propose the **patch self-calibrated tuning** to extract background patches from the new local background similarity. Finally, the softmax function transforms the local class similarity to its probabilistic form, which produces consistent probabilities for background patches across all classes by through the OOD loss.

due to some reasons, *e.g.*, low quality images, the effectiveness of background extraction will be influenced. For example, LoCoOp fully utilizes the local class similarity to extract background information, suffering from performance degradation on samples of inaccurate local class similarity (Miyai et al., 2023a). Second, most existing methods adopt fixed strategies for background extraction, *i.e.*, assigning a fixed task objection for background extraction for all images. This obviously ignores the sample diversity because different samples should have different strategies for background extraction. Such fixed strategies often lead to the inclusion of erroneous background information. For instance, SCT (Yu et al., 2024) employs a fixed top-K strategy that neglects the variability of the background patches across samples, thereby easily extracting erroneous background information.

To address the above issues, in this paper, we propose a new FG-BG decomposition method for FS-OOD detection, referred to as **Mambo**, as illustrated in Figure 1, including a background prompt, a local similarity refinement and a patch self-calibrated tuning. Specifically, background prompt is used to obtain the local background similarity, which is further updated based on the local class similarity, *i.e.*, the local similarity refinement for short. As a result, we obtain a more accurate local background similarity. Moreover, both the local background similarity and the local class similarity are used to extract background information. Hence, our method alleviates the dependence of the local class similarity for the background extraction to explore the first issue in previous methods. In addition, we propose the patch self-calibrated tuning to dynamically extract different numbers of background patches for different images based on the predicted probability of the ground-truth class, effectively reducing the inclusion of erroneous background information, thereby exploring the second issues in previous methods.

Compared to previous methods, the main contributions of our method are listed as follows:

- We propose a novel FS-OOD detection method, where shared background information is learned from ID samples via background prompt. As a result, our method alleviates the dependence of local class similarity and achieves robust FS-OOD detection.

- We integrate the local similarity refinement with patch self-calibrated tuning to effectively leverage the local class similarity and dynamically adjust the background extraction strategy, thereby further enhancing FS-OOD detection performance.

## 2 METHODOLOGY

### 2.1 PRELIMINARY

**Out-of-Distribution (OOD) Detection with VLMs.** VLMs-based OOD detection is designed to train a model using in-distribution (ID) samples specified by the downstream task during the training phase. Its goal is to enabl the model to effectively recognize OOD samples during the testing phase (Esmaeilpour et al., 2022; Ming et al., 2022a; Koner et al., 2021). Formally, we define the ID training distribution as $\mathcal{D}_{\text{train}}^{\text{id}}$ over pairs of input ID images $\boldsymbol{x}^{\text{id}}$ and labels $y^{\text{id}} \in \mathcal{Y}^{\text{id}}$, where $\mathcal{Y}^{\text{id}} = \{1, \cdots, M\}$ denotes the label space for ID classes. During the testing phase, we consider two disjoint distributions, *i.e.*, the ID test distribution $\mathcal{D}_{\text{test}}^{\text{id}}$ and the OOD test distribution $\mathcal{D}_{\text{test}}^{\text{ood}}$, which consists of the input OOD images $\boldsymbol{x}^{\text{ood}}$ and the labels $y^{\text{ood}} \in \mathcal{Y}^{\text{ood}}$ ( $\mathcal{Y}^{\text{ood}} \cap \mathcal{Y}^{\text{id}} = \emptyset$). A discriminant function paradigm commonly used to recognize ID samples from OOD samples is listed as follows:

$$D(\boldsymbol{x}) = \begin{cases} \text{ID}, & S(\boldsymbol{x}) \geq \gamma \\ \text{OOD}, & S(\boldsymbol{x}) < \gamma \end{cases}, \tag{1}$$

where $S(\boldsymbol{x})$ is a scoring function and $\gamma$ is a threshold.

**Prompt Learning with CLIP.** Most of existing FS-OOD detection methods based on prompt learning are trained on learnable text context vectors of CoOp (Zhou et al., 2022c). Specifically, given an ID image $\boldsymbol{x}^{\text{id}}$ and the corresponding ID class label $y^{\text{id}}$, its global image features are obtained by the frozen image encoder $\mathcal{V}(\cdot)$ of CLIP (Radford et al., 2021). After that, $N$ learnable context vectors $\boldsymbol{W} = \{\boldsymbol{w_1}, \boldsymbol{w_2}, \cdots, \boldsymbol{w_N}\}, \boldsymbol{W} \in \mathbb{R}^{N \times d}$ are concatenated to obtain the ID class word features, *i.e.*, $\boldsymbol{C} = \{\boldsymbol{c_1}, \boldsymbol{c_2}, \cdots, \boldsymbol{c_M}\}$, where $\boldsymbol{C} \in \mathbb{R}^{M \times d}$ is the class prompt and $d$ is the feature dimension of CLIP. For instance, the class prompt for class $m$ is denoted as $\boldsymbol{t_m} = \{\boldsymbol{w_1}, \boldsymbol{w_2}, \cdots, \boldsymbol{w_N}, \boldsymbol{c_m}\}, \boldsymbol{t_m} \in \mathbb{R}^{(N+1) \times d}$ and $\boldsymbol{w} = \left\{\boldsymbol{w_n}|_{n=1}^{N}\right\}$ are $N$ learnable context vectors. Next, class text features are obtained by the frozen text encoder $\mathcal{T}(\cdot)$. The final prediction probability of ID class is calculated as follows:

$$p\left(y = i\middle|\boldsymbol{x}^{\text{id}}\right) = \frac{\exp\left(\cos\left(\mathcal{V}\left(\boldsymbol{x}^{\text{id}}\right), \mathcal{T}\left(\boldsymbol{t_i}\right)\right)/\tau\right)}{\sum_{j=1}^{M} \exp\left(\cos\left(\mathcal{V}\left(\boldsymbol{x}^{\text{id}}\right), \mathcal{T}\left(\boldsymbol{t_j}\right)\right)/\tau\right)}, \tag{2}$$

where $\cos(\cdot, \cdot)$ is the cosine similarity operation and $\tau$ is the temperature coefficient.

### 2.2 MOTIVATION

Among CLIP-based prompt learning methods, the foreground-background (FG-BG) decomposition method represents a particularly advanced direction (Ming et al., 2022b; Miyai et al., 2023a; Yu et al., 2024). It focuses on first identifying ID-irrelevant regions using local image features and then explicitly pushing them away from the corresponding ID text features in the feature space. Specifically, given an image, previous FG-BG decomposition methods (Zhou et al., 2022a; Sun et al., 2022; Miyai et al., 2023b) encode this image by the image encoder of CLIP to obtain the global image features $\boldsymbol{f}_{\text{global}}^{\text{id}}$ (*i.e.*, $\boldsymbol{f}_{\text{global}}^{\text{id}} = \mathcal{V}\left(\boldsymbol{x}^{\text{id}}\right)$) for the ID samples, as well as encode each patch token of the image to obtain its local image features by:

$$\boldsymbol{f_i^{\text{id}}} = \text{Proj}_{v \to t}(v(\boldsymbol{l_i^{\text{id}}})), \quad \boldsymbol{f_i^{\text{id}}}, \boldsymbol{l_i^{\text{id}}} \in \mathbb{R}^d, \tag{3}$$

where $\boldsymbol{f_i^{\text{id}}}$ denotes the local image features of the $i$-th patch, $\boldsymbol{l_i^{\text{id}}}$ represents the corresponding feature in the feature map, $v(\cdot)$ is the value projection, and $\text{Proj}_{v \to t}(\cdot)$ denotes the projection operation. After that, previous methods compute the cosine similarity between the class text features (*i.e.*, $\boldsymbol{g_j} = \mathcal{T}(\boldsymbol{t_j})$) and the local image features is regarded as the local class similarity:

$$\boldsymbol{s_i^{\text{class}}}(y = j) = \cos(\boldsymbol{f_i^{\text{id}}}, \boldsymbol{g_j}), \tag{4}$$

where $\boldsymbol{s_i^{\text{class}}}(y = j)$ is the similarity of the $i$-th patch to the $j$-th ID class. Considering that containing rich foreground semantic information can be used to separate local image features, the local class similarity is typically used as the measurement, and is transformed to its probabilistic form by the softmax function. Specifically, they compute the prediction probability of the $n$-th patch by:

$$p_n\left(y = i\middle|\boldsymbol{x}^{\text{id}}\right) = \frac{\exp\left(\cos\left(\boldsymbol{f_n^{\text{id}}}, \boldsymbol{g_i}\right)/\tau\right)}{\sum_{j=1}^{M} \exp\left(\cos\left(\boldsymbol{f_n^{\text{id}}}, \boldsymbol{g_j}\right)/\tau\right)}. \tag{5}$$

Next, a fixed top-K threshold is used to extract background regions $J$ by:

$$J = \left\{ i \in P : \text{rank}\left(p_i(y = y^{\text{id}} \mid \boldsymbol{x}^{\text{id}})\right) > K \right\}, \quad P = \{0, 1, 2, \ldots, H \times W - 1\}, \quad (6)$$

where $P$ is the patch set, $H$ and $W$ denote the height and width of the image feature map. Finally, the entropy function is regarded as the OOD loss function for the OOD regularization, *i.e.*,

$$\mathcal{L}_{\text{OOD}} = -H(p_k), \quad k \in J, \quad (7)$$

where $H(\cdot)$ denotes the entropy function. The optimization of Eq. (7) suppresses semantic information in class text features unrelated to ID classes, thereby enhancing the discriminative ability of the model for OOD samples.

Based on the above analysis, the local class similarity in Eq. (4) is used to conduct background extraction in Eq. (6) as well as the OOD detection in Eq. (7). Hence, previous methods depend heavily on the local class similarity. If it is not accurate, both the background extraction and the effectiveness of OOD detection will be influenced. Furthermore, Eq. (6) makes background extraction with a fixed top-K strategy, ignoring the image diversity. As a result, erroneous background information is easily introduced and the effectiveness of FS-OOD detection is influenced. To address these limitations, we design a new background extraction module including a background prompt, a local similarity refinement and a patch self-calibrated tuning, for FS-OOD detection. Specifically, background prompt in Section 2.3.1 and the local similarity refinement in Section 2.3.2 are used address the first limitation, and the patch calibrated tuning in Section 2.3.3 tackles the second limitation.

## 2.3 TRAINING PHASE OF OUT-OF-DISTRIBUTION DETECTION

### 2.3.1 BACKGROUND PROMPT

**Background Prompt Learning.** Previous FG-BG decomposition methods has been demonstrated to rely heavily on the local class similarity. To address this issue, we propose to learn a background prompt to alleviate the dependence of the local class similarity for background extraction.

Intuitively, we can learn a background prompt from either the image encoder or the text encoder of CLIP. If we use the image encoder to learn the background prompt, our model need more parameters because every patch should learn a prompt. In contrast, if we use the text encoder to learn the background prompt, we only learn one prompt to extract the background information, resulting in less parameters. After employing the text encoder to learn the background prompt, we further propose to learn the background prompt rather than using the class prompt, because the class prompt captures foreground semantics and cannot accurately represent background regions. The background prompt is expected to independently capture background semantic information from different samples, thereby enabling reliable background extraction. Specifically, we define a background prompt as:

$$\boldsymbol{t^b} = \{\boldsymbol{b_1}, \boldsymbol{b_2}, \cdots, \boldsymbol{b_L}\} \in \mathbb{R}^{L \times d}, \quad (8)$$

where each $\boldsymbol{b_i}$ is a learnable token of dimension $d$ and $L$ is the length of background prompt. We then feed the $\boldsymbol{t^b}$ into the the frozen text encoder of CLIP to obtain background text feature of the image:

$$\boldsymbol{g^b} = \mathcal{T}(\boldsymbol{t^b}), \quad \boldsymbol{g^b} \in \mathbb{R}^d. \quad (9)$$

The text encoder maps prompts including the background prompt into a rich semantic space, where a background prompt naturally corresponds to background-related concepts. Hence, background text feature naturally contains semantic background information. This allows the background text feature to serve as reliable references for identifying background regions.

**Local Background Similarity.** The cosine similarity between local image features and background text feature reflects the degree of alignment between individual patches and the background semantics, local background similarity for short. Compared to the local class similarity, the local background similarity is more robust because local background similarity cannot be influenced by the low-accuracy samples. Therefore, we compute it for background extraction.

To do this, after obtaining local image features and background text feature via Eqs. (3) and (9), we define the cosine similarity between the $i$-th patch and background text feature as local background similarity of the corresponding image patch:

$$\boldsymbol{s_i} = \cos(\boldsymbol{f_i^{\text{id}}}, \boldsymbol{g^b}). \quad (10)$$

We utilize the background prompt to obtain the local background similarity. By the local background similarity, we alleviate dependence on local class similarity because the background semantics are independently obtained through the background prompt, without relying on class-related information. As a result, it effectively alleviates the poor performance of previous FG-BG methods on low-accuracy samples and contributes to explore the first issue in previous method.

### 2.3.2 LOCAL SIMILARITY REFINEMENT

In Section 2.3.1, we design the local background similarity to alleviate reliance on the local class similarity. We have the following observations. First, the local class similarity contains rich semantic image information with great potential for background extraction, but it has not been sufficiently and effectively exploited for the learning of local background similarity, which may still contain inaccurate background information. This limitation may cause the model to fail in accurately exploiting local semantic information, which constitutes a key limitation of previous FG-BG methods. Second, since the background prompt does not learn sample-specific semantic information, it may either introduce noise or fail to capture the background information of particular samples. Therefore, it is possible to refine the local background similarity by foreground semantic information in the local class similarity.

To refine the local background similarity, an intuitive solution is to adaptively utilize the local class similarity based on the predicted probability for the ground-truth class. This is because the local class similarity may be unreliable for samples with low predicted probability, so that its contribution needs to be flexibly adjusted according to the prediction confidence to improve the reliability of background extraction. Therefore, we propose to utilize the local class similarity to refine the local background similarity in Eq. (10) by:

$$\boldsymbol{s_i} = \boldsymbol{s_i} \times \left[ (1 - p(y^{\mathrm{id}}|\boldsymbol{x}^{\mathrm{id}})) + p(y^{\mathrm{id}}|\boldsymbol{x}^{\mathrm{id}}) \cdot \Delta_i \right], \text{ where } \Delta_i = \frac{\max_{j \in P} \boldsymbol{s_j^{\mathrm{class}}} - \boldsymbol{s_i^{\mathrm{class}}}}{\max_{j \in P} \boldsymbol{s_j^{\mathrm{class}}} - \min_{j \in P} \boldsymbol{s_j^{\mathrm{class}}}}, \quad (11)$$

where $\Delta_i$ is the refinement weight factor, $\boldsymbol{s_j^{\mathrm{class}}}$ is the local class similarity of the $j$-th patch. We use Eq. (11) to effectively utilize local class similarity to refine local background similarity, thereby obtaining accurate local background similarity. As a result, our method depends on both the local class similarity and the local background similarity to conduct background extraction. Compared to previous FG-BG decomposition methods (*e.g.*, LoCoOp) relying on the local class similarity only, our method is more flexible and effective.

### 2.3.3 PATCH SELF-CALIBRATED TUNING

Although the local background similarity becomes more accuracy by the local similarity refinement, the background extraction process in previous methods still suffers from another limitation, *i.e.*, the lack of flexibility in determining the number of the background patch extraction according to task objectives. For example, LoCoOp directly adopts a fixed top-K strategy to extract background patches. However, every image obviously has different number of background patches to be extract according to task objectives. Hence, the fixed thresholding strategy fails to adapt task objectives based on the predicted probability for the ground-truth class, leading to overfitting and the extraction of erroneous background patches.

To address this issue, we further design the patch self-calibrated tuning to flexibly extract background patches. To do this, we observe an intuitive perspective as follows. Specifically, if the class prompt fails to capture sufficient semantic information with low-accuracy samples, less background patches should be extracted to encourage learning of ID-relevant feature, thereby improving classification performance. In contrast, if the class prompt already aligns well with high-accuracy samples, more background patches should be extracted to suppress OOD feature, thereby enhancing OOD detection. Based on the above analysis, we propose the patch self-calibrated tuning to dynamically adjust extracted background patches according to the predicted probability for the ground-truth class.

Specifically, we extract background patches from the patch set $P$ by:

$$J = \{ i \in P : \boldsymbol{s}_i > \theta \}, \text{ where } \theta = \bar{s} - \alpha \cdot (2p(y^{\mathrm{id}}|\boldsymbol{x}^{\mathrm{id}}) - 1) \cdot \sigma_s, \quad (12)$$

where $\theta$ is the extraction threshold, $\alpha$ is a hyperparameter to adjust the degree of tuning, $\bar{s}$ denotes the average cosine similarity of all patches, and $\sigma_s$ is the standard deviation of local background

---

**Algorithm 1:** the pipeline of Mambo

---

**Input:** learnable prompt $\boldsymbol{w}$ and $\boldsymbol{t^b}$, fine-tuning epochs $T$, learning rate $\eta$, a training set of ID
data, training batch $M$, extraction patches parameter $\alpha$, regularization weight $\lambda$, text
encoder $\mathcal{T}(\cdot)$, image encoder $\mathcal{V}(\cdot)$;

**Output:** fine-tuned prompt $\boldsymbol{w}$ and $\boldsymbol{t^b}$;

1 **for** $epoch = 1, \ldots, T$ **do**
2     **for** $mini\text{-}batch = 1, \ldots, M$ **do**
3        Sample a mini-batch $\{(x_i^{\mathrm{id}}, y_i^{\mathrm{id}})\}_{i=1}^n$ from the training set;
4        Calculate local class similarity and local background similarity by Eq. (4) and Eq. (10);
5        Sample refine local background similarity by Eq. (11);
6        Sample extract background patches from ID local image feature by Eq. (12);
7        $\boldsymbol{w} \leftarrow \boldsymbol{w} - \eta \nabla_{\boldsymbol{w}} \left[ \frac{1}{n} \sum \mathcal{L}_{\mathrm{CE}} \times (1 - p(y^{\mathrm{id}}|\boldsymbol{x}^{\mathrm{id}})) + \lambda \mathcal{L}_{\mathrm{OOD}} \times p(y^{\mathrm{id}}|\boldsymbol{x}^{\mathrm{id}}) \right]$;
8        $\boldsymbol{t^b} \leftarrow \boldsymbol{t^b} - \eta \nabla_{\boldsymbol{t^b}} \left[ \frac{1}{n} \sum \mathcal{L}_{\mathrm{CE}} \times (1 - p(y^{\mathrm{id}}|\boldsymbol{x}^{\mathrm{id}})) + \lambda \mathcal{L}_{\mathrm{OOD}} \times p(y^{\mathrm{id}}|\boldsymbol{x}^{\mathrm{id}}) \right]$;

---

similarity across all patches. Eq. (12) adaptively extracts the number of the background patches, thereby flexibly adjusting task objectives under different accuracy samples and improving FS-OOD detection performance. After adaptively ranking the patches for the patch set, our method is able to select different numbers of background patches for different images, taking the sample diversity into account. Hence, our method explores the second issue in previous methods by Eq. (12).

### 2.3.4 OBJECTIVE FUNCTION

Previous studies have shown that the class prompt tends to overfit background information in ID samples, and thus leading to low confidence scores for OOD samples during the training phase (Ming et al., 2022b; Miyai et al., 2023a). To mitigate this issue, we follow the regularization strategy in LoCoOp (Miyai et al., 2023a) to consider both Eq. (5) and Eq. (7) as the regularization terms in the objective function of our proposed method. As a result, the overall objective function used to train our framework can be formulated as:

$$\mathcal{L} = \mathbb{E}_{(\boldsymbol{x}^{\mathrm{id}}, y^{\mathrm{id}}) \sim \mathcal{D}_{\mathrm{train}}^{\mathrm{id}}} \left[ \mathcal{L}_{\mathrm{CE}} \times (1 - p(y^{\mathrm{id}}|\boldsymbol{x}^{\mathrm{id}})) + \lambda \mathcal{L}_{\mathrm{OOD}} \times p(y^{\mathrm{id}}|\boldsymbol{x}^{\mathrm{id}}) \right], \tag{13}$$

where $\mathcal{L}_{\mathrm{CE}}$ denotes the cross-entropy loss, $\lambda$ is a balancing hyperparameter, and $p(y^{\mathrm{id}}|\boldsymbol{x}^{\mathrm{id}})$ serves as a modulation factor for self-calibrated tuning (Yu et al., 2024) inspired designed to mitigate the impact of erroneous information by regularizing the loss function.

By optimizing Eq. (13), the proposed method can conduct efficiently OOD detection based on both the local background similarity and the local class similarity. In the objective function, the local background similarity is used to extract background regions, while the local class similarity is employed to compute the OOD loss. We summarize the whole procedure of the proposed Mambo in Algorithm 1.

### 2.4 TESTING PHASE OF OUT-OF-DISTRIBUTION DETECTION

In the testing phase, since our proposed method effectively utilizes local background semantic information for OOD detection, we employ the R-MCM score in (Zeng et al., 2025) to take into account the local OOD semantic information. Specifically, it combines the maximum softmax probability scores for global, local class and local background similarity by:

$$S_{\mathrm{R\text{-}MCM}} = S_{\mathrm{MCM}} + R_q^{\mathrm{mean}} \left\{ \frac{\exp\left(\cos(\boldsymbol{f}_{\boldsymbol{z}}^{\mathrm{id}}, \mathcal{T}(\boldsymbol{t_i}))/\tau\right)}{\sum_{j=1}^{M} \exp\left(\cos(\boldsymbol{f}_{\boldsymbol{z}}^{\mathrm{id}}, \mathcal{T}(\boldsymbol{t_j}))/\tau\right) + \exp\left(\boldsymbol{s_z}/\tau\right)} \right\}, \tag{14}$$

where

$$S_{\mathrm{MCM}} = \max_{m} \frac{\exp\left(\cos(\mathcal{V}(\boldsymbol{x}^{\mathrm{id}}), \mathcal{T}(\boldsymbol{t_m}))/\tau\right)}{\sum_{m'=1}^{M} \exp\left(\cos(\mathcal{V}(\boldsymbol{x}^{\mathrm{id}}), \mathcal{T}(\boldsymbol{t_{m'}}))/\tau\right)}, \tag{15}$$

$R_q^{\text{mean}}$ is the mean of $q$ largest elements in all patches, $\boldsymbol{f}_z^{id}$ is the image feature of the $z$-th patch, and $\tau = 1$ in testing phase. In the testing phase, the method accounts for local background semantic information to more accurately remove its influence, which enlarges the distinction between ID and OOD samples and leads to superior OOD detection results.

## 3 EXPERIMENT

### 3.1 EXPERIMENTAL DETAIL

**Datasets.** To evaluate the effectiveness of our methods, we conduct two kinds of experiments, *i.e.*, OOD detection and near OOD detetion. For OOD detection, we follow the literature (Zeng et al., 2025; Miyai et al., 2023a; Yu et al., 2024; Ming et al., 2022a) to consider two datasets (*i.e.*, ImageNet-1K (Deng et al., 2009) and ImageNet-100 (Ming et al., 2022a)) as ID datasets and follow the literature (Huang & Li, 2021) to consider four datasets (*i.e.*, iNaturalist (Van Horn et al., 2018), SUN (Xiao et al., 2010), Places (Zhou et al., 2017), and Texture (Cimpoi et al., 2014)) as OOD datasets. We also build experiments of OOD detection on OpenOOD benchmark (Zhang et al., 2023). Specifically, we use ImageNet-1K as the ID dataset and datasets (*i.e.*, iNaturalist, Texture, and OpenImage-O (Wang et al., 2022)) as OOD datasets. For near OOD detection, we use ImageNet-1K as the ID dataset and three datasets (*i.e.*, ImageNet-O (Hendrycks et al., 2021), SSB-hard (Vaze et al., 2021) and NINCO (Bitterwolf et al., 2023)) as near OOD datasets. In addition, we use two semantically similar subsets of ImageNet-1K (*i.e.*, ImageNet-10 and ImageNet-20) to evaluate near OOD detection performance (Ming et al., 2022a). More details are provided in Appendix A.3.

**Comparison Methods.** The comparison methods include two zero-shot detection methods (*i.e.*, MCM (Ming et al., 2022a) and GL-MCM (Miyai et al., 2025)), six fine-tuned detection methods (*i.e.*, MSP (Hendrycks & Gimpel, 2017), ODIN (Liang et al., 2018), Energy (Liu et al., 2020), ReAct (Sun et al., 2021), MaxLogit (Hendrycks et al., 2022), and NPOS (Tao et al., 2023)), and four few-shot detection methods (*i.e.*, CoOp (Zhou et al., 2022c), LoCoOp (Miyai et al., 2023a), Local-Prompt (Zeng et al., 2025), and SCT (Yu et al., 2024)). More details of comparison methods can be found in Appendix A.4.1.

**Implementation details.** We follow the literature (Zeng et al., 2025; Miyai et al., 2023a; Yu et al., 2024) to use VIT-B/16 (Dosovitskiy et al., 2020) as the backbone, where we froze the backbone and only train the class prompt and the background prompt. For our method Mambo, we adopt $\lambda = 0.2$ under all few-shot setting. We train the CLIP for 30 epochs with a learning rate of 0.002 and other hyperparameters (*e.g.*, batch size = 32, SGD optimizer and token length of class prompt $N = 16$) are the same as those of CoOp (Zhou et al., 2022c). For all comparison methods, we follow their literature to set parameters so that all of them output their best performance in our experiments. More detail are shown in Appendix A.4.2.

**Evaluation metrics.** The evaluation metrics include the false positive rate of OOD samples when the true positive rate of ID samples is 95% (FPR95) and the area under the receiver operating characteristic curve (AUROC) (Davis & Goadrich, 2006).

### 3.2 MAIN RESULTS

We report the OOD detection results of all methods on four OOD datasets by fixing ID datasets as ImageNet-1K and ImageNet-100, respectively, in Tables 1 and 2 (more results can be seen in Appendix A.5). We also report the results of the OOD detection and the near OOD detection on OpenOOD benchmark of all FS-OOD methods in Table 9 (see Appendix A.5). In addition, we report the near OOD detection results of all FS-OOD methods on datasets ImageNet-10 and ImageNet-20 in Table 10 (see Appendix A.5).

**OOD detection** First, our method achieves the best results, followed by Local-Prompt, SCT, and LoCoOp, CoOp, GL-MCM, NPOS, MCM, MaxLogit, ReAct, Energy, MSP, and ODIN, in terms of all different few-shot settings on all datasets. For instance, our method reduces by 5.86% and 71.95%, respectively, compared to the best comparison method (*i.e.*, LoCoOp) and the worst comparison method (*i.e.*, ODIN) on 1-shot in terms of FPR95 in ImageNet-1K as the ID datasets. The reason may be that our proposed method take into account the local class similarity less and extracts different numbers of background patches for different images. Second, compared to the non FS-OOD methods

Table 1: **Comparison results on ImageNet-1K OOD benchmarks.** All methods are trained on CLIP-ViT-B/16. ↓ indicates that smaller values are better and ↑ indicates that larger values are better. Results marked with † are obtained from (Miyai et al., 2023a) and (Yu et al., 2024). The few-shot detection methods are reported the the mean and standard deviation over several repeats.

| Method | iNaturalist | | SUN | | Places | | Texture | | Average | |
|---|---|---|---|---|---|---|---|---|---|---|
| | FPR95↓ | AUROC↑ | FPR95↓ | AUROC↑ | FPR95↓ | AUROC↑ | FPR95↓ | AUROC↑ | FPR95↓ | AUROC↑ |
| *Zero-shot* | | | | | | | | | | |
| MCM† | 30.94 | 94.61 | 37.67 | 92.56 | 44.76 | 89.76 | 57.91 | 86.10 | 42.82 | 90.76 |
| GL-MCM† | 15.18 | 96.71 | 30.42 | 93.09 | 38.85 | 89.90 | 57.93 | 83.63 | 35.47 | 90.83 |
| *Fine-tuned* | | | | | | | | | | |
| MSP† | 74.57 | 77.74 | 76.95 | 73.97 | 79.72 | 72.18 | 73.66 | 74.84 | 74.98 | 76.22 |
| ODIN† | 98.93 | 57.73 | 88.72 | 78.42 | 87.80 | 76.88 | 85.47 | 71.49 | 90.23 | 71.13 |
| Energy† | 64.98 | 87.18 | 46.42 | 91.17 | 57.40 | 87.33 | 50.39 | 88.22 | 54.80 | 88.48 |
| ReAct† | 65.57 | 86.87 | 46.17 | 91.04 | 56.85 | 87.42 | 49.88 | 88.13 | 54.62 | 88.37 |
| MaxLogit† | 60.88 | 88.03 | 44.83 | 91.16 | 55.54 | 87.45 | 48.72 | 88.63 | 52.49 | 88.82 |
| NPOS† | 16.58 | 96.19 | 43.77 | 90.44 | 45.27 | 89.44 | 46.12 | 88.80 | 37.93 | 91.22 |
| *1-shot* | | | | | | | | | | |
| $\text{CoOp}_{MCM}$ | $43.60^{\pm2.33}$ | $91.65^{\pm1.07}$ | $41.05^{\pm2.35}$ | $91.20^{\pm0.80}$ | $47.50^{\pm1.79}$ | $88.65^{\pm0.62}$ | $47.27^{\pm1.56}$ | $88.59^{\pm0.47}$ | $44.86^{\pm1.19}$ | $90.02^{\pm0.37}$ |
| $\text{CoOp}_{GL}$ | $21.78^{\pm6.64}$ | $95.16^{\pm2.00}$ | $34.96^{\pm1.08}$ | $91.29^{\pm0.34}$ | $42.56^{\pm1.70}$ | $88.67^{\pm0.29}$ | $49.19^{\pm2.96}$ | $85.87^{\pm1.48}$ | $37.12^{\pm2.75}$ | $90.24^{\pm1.01}$ |
| LoCoOp | $22.12^{\pm4.10}$ | $95.43^{\pm0.84}$ | $23.88^{\pm0.87}$ | $\mathbf{94.84}^{\pm0.14}$ | $33.92^{\pm1.70}$ | $91.48^{\pm0.22}$ | $49.15^{\pm4.20}$ | $87.45^{\pm1.56}$ | $32.27^{\pm2.50}$ | $92.30^{\pm0.65}$ |
| Local-Prompt | $26.06^{\pm4.89}$ | $95.34^{\pm0.60}$ | $27.95^{\pm0.03}$ | $94.46^{\pm0.37}$ | $37.58^{\pm1.22}$ | $91.07^{\pm0.11}$ | $\mathbf{44.61}^{\pm4.93}$ | $\mathbf{89.87}^{\pm1.34}$ | $34.05^{\pm2.75}$ | $\mathbf{92.69}^{\pm0.41}$ |
| SCT | $21.37^{\pm11.25}$ | $95.36^{\pm2.50}$ | $26.97^{\pm2.71}$ | $93.56^{\pm1.00}$ | $35.28^{\pm1.92}$ | $90.71^{\pm0.91}$ | $50.59^{\pm1.61}$ | $86.21^{\pm0.74}$ | $33.56^{\pm4.25}$ | $91.46^{\pm1.20}$ |
| Mambo | $\mathbf{19.74}^{\pm1.10}$ | $\mathbf{95.74}^{\pm0.64}$ | $23.41^{\pm1.72}$ | $94.46^{\pm0.44}$ | $\mathbf{31.00}^{\pm1.06}$ | $\mathbf{91.84}^{\pm0.59}$ | $47.36^{\pm1.61}$ | $86.92^{\pm0.49}$ | $\mathbf{30.38}^{\pm0.48}$ | $92.24^{\pm0.36}$ |
| *4-shot* | | | | | | | | | | |
| $\text{CoOp}_{MCM}$ | $33.80^{\pm8.66}$ | $93.05^{\pm1.79}$ | $34.14^{\pm3.89}$ | $92.59^{\pm0.69}$ | $41.58^{\pm2.56}$ | $89.74^{\pm0.53}$ | $47.41^{\pm1.26}$ | $88.98^{\pm0.31}$ | $39.23^{\pm2.62}$ | $91.09^{\pm0.62}$ |
| $\text{CoOp}_{GL}$ | $17.13^{\pm1.68}$ | $96.15^{\pm0.31}$ | $27.08^{\pm1.48}$ | $93.20^{\pm0.40}$ | $35.46^{\pm0.09}$ | $90.29^{\pm0.31}$ | $51.37^{\pm0.54}$ | $85.29^{\pm0.55}$ | $32.76^{\pm0.22}$ | $91.23^{\pm0.26}$ |
| LoCoOp | $16.35^{\pm2.67}$ | $96.46^{\pm0.46}$ | $22.79^{\pm1.91}$ | $95.00^{\pm0.07}$ | $32.17^{\pm1.34}$ | $91.87^{\pm0.28}$ | $46.33^{\pm3.08}$ | $88.87^{\pm0.31}$ | $29.41^{\pm1.56}$ | $93.05^{\pm0.15}$ |
| Local-Prompt | $13.14^{\pm2.49}$ | $97.28^{\pm0.48}$ | $22.41^{\pm0.60}$ | $95.02^{\pm0.18}$ | $32.48^{\pm0.27}$ | $91.98^{\pm0.18}$ | $\mathbf{41.79}^{\pm0.23}$ | $\mathbf{90.57}^{\pm0.09}$ | $27.45^{\pm0.53}$ | $\mathbf{93.71}^{\pm0.13}$ |
| SCT | $12.92^{\pm2.61}$ | $97.21^{\pm0.46}$ | $21.75^{\pm0.62}$ | $95.08^{\pm0.16}$ | $31.60^{\pm0.87}$ | $91.90^{\pm0.40}$ | $46.30^{\pm1.25}$ | $87.97^{\pm0.16}$ | $28.14^{\pm0.42}$ | $93.04^{\pm0.19}$ |
| Mambo | $\mathbf{11.54}^{\pm2.41}$ | $\mathbf{97.41}^{\pm0.51}$ | $\mathbf{20.27}^{\pm0.89}$ | $\mathbf{95.09}^{\pm0.41}$ | $\mathbf{27.77}^{\pm0.60}$ | $\mathbf{92.89}^{\pm0.24}$ | $47.01^{\pm3.16}$ | $87.89^{\pm0.79}$ | $\mathbf{26.65}^{\pm0.07}$ | $93.32^{\pm0.15}$ |
| *16-shot* | | | | | | | | | | |
| $\text{CoOp}_{MCM}$ | $29.79^{\pm2.55}$ | $93.82^{\pm0.58}$ | $35.34^{\pm1.15}$ | $92.46^{\pm0.42}$ | $42.16^{\pm0.38}$ | $89.94^{\pm0.15}$ | $42.73^{\pm2.55}$ | $90.14^{\pm0.61}$ | $37.50^{\pm0.90}$ | $91.59^{\pm0.03}$ |
| $\text{CoOp}_{GL}$ | $15.55^{\pm1.04}$ | $96.44^{\pm0.33}$ | $29.01^{\pm1.84}$ | $92.42^{\pm0.35}$ | $36.60^{\pm1.68}$ | $90.00^{\pm0.19}$ | $46.47^{\pm0.27}$ | $86.43^{\pm0.64}$ | $31.91^{\pm1.06}$ | $91.32^{\pm0.12}$ |
| LoCoOp | $17.27^{\pm0.84}$ | $96.44^{\pm0.08}$ | $23.28^{\pm0.74}$ | $95.11^{\pm0.16}$ | $32.19^{\pm1.59}$ | $92.05^{\pm0.23}$ | $43.32^{\pm3.08}$ | $89.41^{\pm0.79}$ | $29.01^{\pm1.48}$ | $93.26^{\pm0.27}$ |
| Local-Prompt | $\mathbf{9.01}^{\pm0.56}$ | $\mathbf{98.11}^{\pm0.11}$ | $22.47^{\pm0.08}$ | $95.24^{\pm0.04}$ | $31.26^{\pm0.59}$ | $92.52^{\pm0.07}$ | $\mathbf{36.77}^{\pm0.95}$ | $\mathbf{91.74}^{\pm0.03}$ | $\mathbf{24.88}^{\pm0.24}$ | $\mathbf{94.40}^{\pm0.01}$ |
| SCT | $15.33^{\pm2.24}$ | $96.77^{\pm0.28}$ | $20.93^{\pm1.51}$ | $95.10^{\pm0.29}$ | $29.98^{\pm1.15}$ | $92.19^{\pm0.04}$ | $44.88^{\pm1.70}$ | $88.20^{\pm0.42}$ | $27.78^{\pm1.56}$ | $92.82^{\pm0.48}$ |
| Mambo | $13.10^{\pm1.46}$ | $96.98^{\pm0.49}$ | $\mathbf{18.65}^{\pm0.59}$ | $\mathbf{95.61}^{\pm0.27}$ | $\mathbf{26.65}^{\pm0.12}$ | $\mathbf{93.21}^{\pm0.11}$ | $42.83^{\pm0.47}$ | $88.92^{\pm0.23}$ | $25.31^{\pm0.30}$ | $93.68^{\pm0.20}$ |

Table 2: Experiments on ImageNet-100 as the ID dataset with 4-shot few-shot tuning results.

| Method | iNaturalist | | SUN | | Places | | Texture | | Average | |
|---|---|---|---|---|---|---|---|---|---|---|
| | FPR95↓ | AUROC↑ | FPR95↓ | AUROC↑ | FPR95↓ | AUROC↑ | FPR95↓ | AUROC↑ | FPR95↓ | AUROC↑ |
| LoCoOp (Miyai et al., 2023a) | $9.70^{\pm3.15}$ | $97.98^{\pm0.51}$ | $12.73^{\pm0.52}$ | $97.42^{\pm0.08}$ | $18.46^{\pm1.03}$ | $96.18^{\pm0.18}$ | $25.02^{\pm2.64}$ | $95.08^{\pm0.40}$ | $16.48^{\pm0.90}$ | $96.66^{\pm0.13}$ |
| Local-Prompt (Zeng et al., 2025) | $\mathbf{7.21}^{\pm2.58}$ | $\mathbf{98.38}^{\pm0.31}$ | $15.33^{\pm2.98}$ | $97.27^{\pm0.46}$ | $21.53^{\pm3.77}$ | $96.02^{\pm0.50}$ | $\mathbf{21.03}^{\pm3.58}$ | $\mathbf{96.26}^{\pm0.37}$ | $16.28^{\pm1.22}$ | $96.98^{\pm0.14}$ |
| SCT (Yu et al., 2024) | $12.93^{\pm5.11}$ | $97.58^{\pm0.60}$ | $11.53^{\pm1.62}$ | $97.61^{\pm0.43}$ | $17.85^{\pm0.23}$ | $96.18^{\pm0.15}$ | $27.81^{\pm3.29}$ | $94.64^{\pm0.74}$ | $17.53^{\pm1.44}$ | $96.50^{\pm0.27}$ |
| Mambo | $9.09^{\pm4.51}$ | $98.08^{\pm0.67}$ | $\mathbf{10.74}^{\pm3.22}$ | $\mathbf{97.81}^{\pm0.57}$ | $\mathbf{15.63}^{\pm3.23}$ | $\mathbf{96.77}^{\pm0.55}$ | $23.04^{\pm3.96}$ | $95.63^{\pm0.80}$ | $\mathbf{14.63}^{\pm2.10}$ | $\mathbf{97.07}^{\pm0.38}$ |

(*e.g.*, GL-MCM, NPOS, MCM, MaxLogit, ReAct, Energy, MSP, and ODIN) with the FS-OOD methods (*e.g.*, our proposed Mambo, LoCoOp, SCT, and Local-Prompt), all FS-OOD methods beat other methods. For example, the worst FS-OOD methods (*i.e.*, LoCoOp) reduces by 22.27%, compared to the best non FS-OOD methods (*i.e.*, GL-MCM), in terms of FPR95. The reason is that all FS-OOD methods employ prompt learning for background extraction and others do not train any prompt. This verifies that it is reasonable to use prompt for OOD detection. In particular, our method outperforms all other FS-OOD methods, *i.e.*, Local-Prompt, SCT, and LoCoOp. The reason is that our proposed method ues one more prompt, *i.e.*, the background prompt, to capture background semantic information. Hence, it is feasible to learn the background prompt for FS-OOD detection.

**Near OOD detection** Similar to the FS-OOD results of all methods, our method beats all comparison methods in terms of the near FS-OOD results on all scenarios. For example, our method improves by 4.40% and 6.33%, respectively, compared to the best comparison method (*i.e.*, Local-Prompt) and the worst comparison method (*i.e.*, LoCoOp), in terms of AUROC. Moreover, our method also outperforms all other FS-OOD methods. These results demonstrate that i) it is reasonable to reduce the dependence of the local class similarity and to extract different numbers of background patches for different images and ii) our method is robust because it achieves the best results, compared to all comparison methods, in terms different datasets and different kinds of OOD detection.

## 3.3 ABLATION STUDIES

**Effectiveness of different components.** Our method involves two key components, *i.e.*, the local similarity refinement and the patch self-calibrated tuning. To assess the effectiveness of each component, we report the OOD detection results of all FS-OOD methods on four OOD datasets

Table 3: Experiment on verifying the effectiveness of each component.

| Method | iNaturalist | | SUN | | Places | | Texture | | Average | |
|---|---|---|---|---|---|---|---|---|---|---|
| | FPR95↓ | AUROC↑ | FPR95↓ | AUROC↑ | FPR95↓ | AUROC↑ | FPR95↓ | AUROC↑ | FPR95↓ | AUROC↑ |
| Baseline | $10.42^{\pm1.43}$ | $97.72^{\pm0.34}$ | $23.89^{\pm2.11}$ | $94.34^{\pm0.29}$ | $31.46^{\pm1.70}$ | $91.81^{\pm0.40}$ | $44.44^{\pm2.05}$ | $88.11^{\pm0.84}$ | $27.55^{\pm0.89}$ | $92.99^{\pm0.14}$ |
| w/ local similarity refinement | $16.41^{\pm0.21}$ | $96.29^{\pm0.10}$ | $20.01^{\pm0.65}$ | $95.26^{\pm0.22}$ | $28.66^{\pm1.26}$ | $92.65^{\pm0.28}$ | $43.14^{\pm0.80}$ | $88.68^{\pm0.41}$ | $27.06^{\pm0.70}$ | $93.22^{\pm0.13}$ |
| w/ patch self-calibrated tuning | $11.99^{\pm2.89}$ | $97.40^{\pm0.63}$ | $24.29^{\pm0.17}$ | $94.03^{\pm0.26}$ | $31.73^{\pm0.78}$ | $91.75^{\pm0.17}$ | $\mathbf{41.77}^{\pm0.95}$ | $\mathbf{89.49}^{\pm0.07}$ | $27.44^{\pm0.80}$ | $93.17^{\pm0.15}$ |
| Mambo | $13.10^{\pm1.46}$ | $96.98^{\pm0.49}$ | $\mathbf{18.65}^{\pm0.59}$ | $\mathbf{95.61}^{\pm0.27}$ | $\mathbf{26.65}^{\pm0.12}$ | $\mathbf{93.21}^{\pm0.11}$ | $42.83^{\pm0.47}$ | $88.92^{\pm0.23}$ | $\mathbf{25.31}^{\pm0.30}$ | $\mathbf{93.68}^{\pm0.20}$ |

Figure 2: Visualization of extracted background patches: our method (left) and SCT (right).

using ImageNet-1K as the ID dataset in Table 3. First, the methods with only component (*i.e.*, w/ local similarity refinement and w/ patch self-calibrated tuning) are better than Baseline with only background prompt. For example, w/ local similarity refinement and w/ patch self-calibrated tuning reduce by 1.78% and 0.40%, respectively, compared to Baseline, in terms of FPR95. This implies that each component is useful for OOD detection. More specifically, the similarity refinement provides more reliable local background semantic information by flexibly and effectively utilizing the local class similarity to refine the local background similarity while the patch self-calibrated tuning prevents model overfitting and avoids extracting erroneous background information by introducing a flexible background extraction strategy. Moreover, compared w/ local similarity refinement with w/ patch self-calibrated tuning, w/ patch self-calibrated tuning outperforms baseline a little bit. The reason may be that noise in the local background similarity may interfere with background extraction. Second, Mambo beats all methods because it considers both the local similarity refinement and the patch self-calibrated tuning. This demonstrates again that it is necessary to consider both of them for FS-OOD detection because they are able to conduct effective background extraction.

**Parameter count.** To verify whether Mambo can achieve good performance without introducing additional trainable parameters. we set both the token length of the class prompt (*i.e.*, $N$) and the background prompt length (*i.e.*, $L$) to 8, keeping the number of learnable parameters the same as previous OOD detection methods based on FG-BG decomposition. We report the results of our method on the ImageNet-1K benchmark in Appendix A.6. The results indicate that even with the same number of parameters, our method still outperforms previous FG-BG decomposition methods fo OOD detection methods. For instance, our method improves by 4.75% , respectively, compared to the best FG-BG decomposition method (*i.e.*, SCT), in terms of FPR95. This reflects the effectiveness and efficiency of our method.

**CLIP Architecture.** To validate the effectiveness of our method, we conducted experiments across multiple CLIP backbone architectures and compared our method with the best FG–BG decomposition method, SCT. The results are shown in Table 4, demonstrate that our method consistently achieves superior performance across all architectures, indicating that Mambo outperforms the best FG–BG decomposition methods regardless of the CLIP backbone. Notably, we used the same hyperparameters for all architectures, which further demonstrates the robustness of Mambo's hyperparameters across different VLM architectures.

## 3.4 VISUALIZATION OF BACKGROUND PATCHES

**Visualization of background patches.** We report the visualization results of our proposed method and SCT in terms of background patches in Figure 2, as well as background prompt, distribution map, refinement, failure, difficult foreground samples, more visualization of background concept, and validation of motivation in Appendix A.7. Experimental results demonstrate that our method accurately extracts effective background patches. For instance, On low-accuracy sample (*e.g.*, scarf), SCT retains a significant amount of background information unrelated to foreground. In contrast, our method extracts background information more accurately, leading to improve the FS-OOD detection

Table 4: Experiments on ImageNet-1K as the ID dataset with different CLIP architectures.

| Method | iNaturalist | | SUN | | Places | | Texture | | Average | |
|---|---|---|---|---|---|---|---|---|---|---|
| | FPR95↓ | AUROC↑ | FPR95↓ | AUROC↑ | FPR95↓ | AUROC↑ | FPR95↓ | AUROC↑ | FPR95↓ | AUROC↑ |
| ViT-B/32 | | | | | | | | | | |
| SCT (Yu et al., 2024) | $28.46^{\pm4.89}$ | $94.83^{\pm0.71}$ | $35.18^{\pm3.84}$ | $92.90^{\pm0.54}$ | $41.25^{\pm2.84}$ | $90.62^{\pm0.44}$ | $48.15^{\pm2.62}$ | $88.32^{\pm0.76}$ | $38.26^{\pm2.71}$ | $91.67^{\pm0.34}$ |
| Mambo | $\mathbf{26.75}^{\pm2.02}$ | $\mathbf{95.02}^{\pm0.38}$ | $\mathbf{33.80}^{\pm1.43}$ | $\mathbf{93.22}^{\pm0.25}$ | $\mathbf{40.45}^{\pm1.35}$ | $\mathbf{90.75}^{\pm0.26}$ | $\mathbf{46.73}^{\pm0.45}$ | $\mathbf{88.46}^{\pm0.07}$ | $\mathbf{36.93}^{\pm1.03}$ | $\mathbf{91.87}^{\pm0.15}$ |
| ViT-B/16 | | | | | | | | | | |
| SCT (Yu et al., 2024) | $15.33^{\pm2.24}$ | $96.77^{\pm0.28}$ | $20.93^{\pm1.51}$ | $95.10^{\pm0.29}$ | $29.98^{\pm1.15}$ | $92.19^{\pm0.04}$ | $44.88^{\pm1.70}$ | $88.20^{\pm0.42}$ | $27.78^{\pm1.56}$ | $92.82^{\pm0.48}$ |
| Mambo | $\mathbf{13.10}^{\pm1.46}$ | $\mathbf{96.98}^{\pm0.49}$ | $\mathbf{18.65}^{\pm0.59}$ | $\mathbf{95.61}^{\pm0.27}$ | $\mathbf{26.65}^{\pm0.12}$ | $\mathbf{93.21}^{\pm0.11}$ | $\mathbf{42.83}^{\pm0.47}$ | $\mathbf{88.92}^{\pm0.23}$ | $\mathbf{25.31}^{\pm0.30}$ | $\mathbf{93.68}^{\pm0.20}$ |
| ViT-L/14 | | | | | | | | | | |
| SCT (Yu et al., 2024) | $31.95^{\pm1.93}$ | $93.44^{\pm0.46}$ | $30.71^{\pm0.80}$ | $93.01^{\pm0.62}$ | $35.34^{\pm1.46}$ | $91.09^{\pm0.50}$ | $49.62^{\pm1.61}$ | $85.68^{\pm0.37}$ | $36.91^{\pm0.77}$ | $90.81^{\pm0.27}$ |
| Mambo | $\mathbf{28.21}^{\pm1.80}$ | $\mathbf{94.44}^{\pm0.39}$ | $\mathbf{29.88}^{\pm1.60}$ | $\mathbf{93.38}^{\pm0.23}$ | $\mathbf{35.29}^{\pm1.23}$ | $\mathbf{91.41}^{\pm0.07}$ | $\mathbf{47.68}^{\pm0.77}$ | $\mathbf{87.36}^{\pm0.39}$ | $\mathbf{35.27}^{\pm0.52}$ | $\mathbf{91.65}^{\pm0.15}$ |

performance. Moreover, on high-accuracy samples (*e.g.*, junco), our method achieves superior background extraction performance. Hence, the visualization results demonstrate both the robustness and the effectiveness of our proposed method.

### 3.5 HYPERPARAMETER SENSITIVITY ANALYSIS

Our proposed method involves two hyperparameters, *i.e.*, $\lambda$ in Eq. (13) and $\alpha$ in Eq. (12). We report the sensitivity analysis of the hyperparameters $\lambda$ and $\alpha$ on the ImageNet-1K benchmark in Figure 9 (see Appendix A.7 for details) by setting $\lambda \in \{0.1, 0.2, ..., 0.4\}$ and $\alpha \in \{0.5, 1, 1.5, ..., 2.5\}$. Beyond such ranges, the model performance is very bad, making it difficult to provide representative analytical conclusions. The experiments demonstrate that the performance of our method easily maintains stable within a certain range of the hyperparameters $\lambda$ and $\alpha$. For example, our method still yields performance nearly equivalent to the best results achieved with $\lambda \in \{0.2, 0.3\}$ and $\alpha \in \{1, 2\}$. This indicates that the hyperparameters in our method is easily to be tuned.

## 4 CONCLUSION

In this paper, we proposed a new OOD detection framework guided by background prompt. Specifically, we first learn a background prompt to guide background extraction from ID samples, and then design the local similarity to exclusively rely on the local class similarity. We further investigate the patch self-calibrated tuning to control the number of extracted background patches. Experimental results demonstrated the effectiveness of our method. In this paper, we explore the FS-OOD detection of the classification task in 2D natural images, but FS-OOD detection is urgently needed in open-world scenarios such as autonomous driving (Filos et al., 2020) and medical imaging. While existing methods can't achieve FS-OOD detection across domains simultaneously, we believe Mambo is broadly applicable in domains with background information and will pursue it as future work.

## 5 REPRODUCIBILITY STATEMENT

The experimental setups are provided in Section 3.1 and Appendix A.4.2 to ensure the result reproducibility. In addition, the source code will be released at `https://anonymous.4open.science/r/Mambo-CBC1`.

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

# A APPENDIX

## A.1 THE USE OF LARGE LANGUAGE MODELS

The use of large language models in this work includes correcting grammatical errors and polishing the manuscript, as well as providing assistance in writing the experimental code. The models were not involved in the design of the methodology, experiments, analysis, or the generation of any research content.

## A.2 RELATED WORK

**Prompt learning for vision-language models (VLMs).** Pre-trained VLMs demonstrate exceptional generalization capabilities when trained on large-scale datasets such as ImageNet (Deng et al., 2009). Fine-tuning is often required when using pre-trained VLMs in downstream tasks. However, full fine-tuning is not only time-consuming but also disrupt the original feature space of the pre-trained model. Therefore, efficient transfer learning (ETL) has become a focal point of research in recent years. ETL can be divided into two groups: adapter tuning (Gao et al., 2024b; Zhang et al., 2022) and prompt learning (Zhou et al., 2022c;b; Ghosal et al., 2024; Gao et al., 2024a; Khattak et al., 2023; Chen et al., 2023; Yu et al., 2023). The prompt learning tunes the model by training a small set of prompts. For instance, CoOp (Zhou et al., 2022c) replaces fixed textual templates with a set of learnable context vectors, allowing pre-trained VLMs to adapt to downstream tasks without relying on manually crafted prompt. CoCoOp (Zhou et al., 2022b) further improves transfer performance by transforming image feature into conditional vectors to optimize the generated contexts. IntCoOp (Ghosal et al., 2024) strengthens the recognition capabilities of VLMs on downstream tasks by incorporating attribute information into text prompt. LAMM (Gao et al., 2024a) narrows the gap between pre-trained and downstream class labels by aligning learnable label embeddings. While prompt learning effectively adapts models to downstream tasks, these methods are not designed for FS-OOD detection. As a result, they do not achieve strong FS-OOD detection performance when used directly.

**OOD detection with VLMs.** Based on the powerful pre-trained knowledge of VLMs, it is able to capture more accurate semantic information. Therefore, many studies have used VLMs for OOD detection tasks. MCM (Ming et al., 2022a) implements CLIP-based zero-shot OOD detection using maximum softmax probability (Hendrycks & Gimpel, 2017). LoCoOp (Miyai et al., 2023a) uses OOD regularization to extract ID-irrelevant regions from local features to regularize class prompt. CLIPN (Wang et al., 2023) introduces a standalone negative text encoder to effectively capture negative semantic introduction in ID samples. LSN (Nie et al., 2024) uses both positive and negative prompt to simultaneously measure the similarity of samples to ID classes. NegLabel (Jiang et al., 2024) learns negative labels from an extensive corpus to identify OOD samples. ID-like (Bai et al., 2024) learns outliers in the space around ID samples by downsampling to distinguish OOD samples. SCT (Yu et al., 2024) utilizes self-calibrating tuning flexible balancing of importance across tasks to reduce the impact of erroneous background information. NegPrompt (Li et al., 2024) utilizes negative prompt to improve open vocabulary OOD detection performance. CoVer (Zhang et al., 2024) uses common corruptions in the input space to upsize the confidence difference between ID and OOD samples. CLIP Surgery (Li et al., 2025) greatly improves the interpretability of OOD detection by performing surgical-level modifications on the architecture and features. CLIP-OS (Sun et al., 2024) leverages the property that foreground patches in the original CLIP have lower similarity to the true class, thereby enhancing the discrimination between foreground and background boundaries.

## A.3 DATASET DETAILS

**ImageNet-100.** ImageNet-100 is a subset of ImageNet-1K, which contains 100 classes as MCM (Ming et al., 2022a) to get a comparison with previous VLMs-based OOD detection methods. The specific class names and numbers are shown on Table 5.

**ImageNet-20.** ImageNet-20 is a subset of ImageNet-1K, which contains 20 classes. The specific class names and numbers are shown on Table 6.

**ImageNet-10.** ImageNet-10 is a subset of ImageNet-1K, which contains 10 classes. The specific class names and numbers are shown on Table 7.

Table 5: ImageNet-100 specific details

| n01498041 | Stingray | n01518878 | Ostrich | n01580077 | Jay | n01601694 | American dipper | n01632458 | Spotted salamander |
|---|---|---|---|---|---|---|---|---|---|
| n01689811 | Alligator lizard | n01695060 | Komodo dragon | n01775062 | Wolf spider | n01817953 | African grey parrot | n01843065 | Jacamar |
| n01855032 | Red-breasted merganser | n01871265 | Tusker | n01910747 | Jellyfish | n01917289 | Brain coral | n01944390 | Snail |
| n02002556 | White stork | n02033041 | Dowitcher | n02058221 | Albatross | n02088364 | Beagle | n02091635 | Otterhound |
| n02095570 | Lakeland Terrier | n02097130 | Giant Schnauzer | n02102318 | Cocker Spaniel | n02105412 | Australian Kelpie | n02107312 | Miniature Pinscher |
| n02111889 | Samoyed | n02113186 | Cardigan Welsh Corgi | n02113799 | Standard Poodle | n02124075 | Egyptian Mau | n02128757 | Snow leopard |
| n02128925 | Jaguar | n02134084 | Polar bear | n02233338 | Cockroach | n02326432 | Hare | n02480495 | Orangutan |
| n02483362 | Gibbon | n02484975 | Guenon | n02488702 | Black-and-white colobus | n02493793 | Geoffroy's spider monkey | n02808304 | Bath towel |
| n02825657 | Bell tower | n02843684 | Birdhouse | n02871525 | Bookstore | n02971356 | Cardboard box / carton | n03000684 | Chainsaw |
| n03016953 | Chiffonier | n03110669 | Cornet | n03125729 | Cradle | n03127925 | Crate | n03133878 | Crock Pot |
| n03180011 | Desktop computer | n03187595 | Rotary dial telephone | n03218198 | Dog sled | n03272562 | Electric locomotive | n03355925 | Flagpole |
| n03388549 | Four-poster bed | n03394916 | French horn | n03400231 | Frying pan | n03404251 | Fur coat | n03425413 | Gas pump |
| n03447721 | Gong | n03457902 | Greenhouse | n03594945 | Jeep | n03633091 | Ladle | n03666591 | Lighter |
| n03710721 | One-piece bathing suit | n03721384 | Marimba | n03840681 | Ocarina | n03866082 | Overskirt | n03877845 | Palace |
| n03887697 | Paper towel | n03895866 | Railroad car | n03908714 | Pencil sharpener | n03929855 | Pickelhaube | n03933933 | Pier |
| n03935335 | Piggy bank | n03982430 | Pool table | n03995372 | Power drill | n04037443 | Race car | n04041544 | Radio |
| n04090263 | Rifle | n04136333 | Sarong | n04147183 | Schooner | n04179913 | Sewing machine | n04239074 | Sliding door |
| n04356056 | Sunglasses | n04371430 | Swim trunks / shorts | n04376876 | Syringe | n04418357 | Front curtain | n04461696 | Tow truck |
| n04483307 | Trimaran | n04550184 | Wardrobe | n04562935 | Water tower | n04606251 | Shipwreck | n06785654 | Crossword |
| n07614500 | Ice cream | n07714571 | Cabbage | n09399592 | Promontory | n09835506 | Baseball player | n13052670 | Hen of the woods mushroom |

Table 6: ImageNet-20 specific details

| n01630670 | Smooth newt | n01631663 | Newt | n01632458 | Spotted salamander | n01693334 | European green lizard | n01697457 | Nile crocodile |
|---|---|---|---|---|---|---|---|---|---|
| n02114367 | Grey wolf | n02120079 | Arctic fox | n02132136 | Brown bear | n02317335 | Starfish | n02391049 | Zebra |
| n02782093 | Balloon | n02917067 | High-speed train | n02951358 | Canoe | n03773504 | Missile | n03785016 | Moped |
| n04147183 | Schooner | n04252077 | Snowmobile | n04266014 | Space shuttle | n04310018 | Steam locomotive | n04389033 | Tank |

**iNaturalist.** iNaturalist (Van Horn et al., 2018) is a large nature dataset which contains 85000+ samples in 5000 classes. Following previous methods (Huang & Li, 2021; Miyai et al., 2023a; Wang et al., 2023), we use a subset of it which contains 10,000 samples in 110 classes as one of OOD datasets.

**SUN.** SUN (Xiao et al., 2010) is a large scene dataset which contains 130000 samples in 397 classes. Following previous methods (Huang & Li, 2021; Miyai et al., 2023a; Wang et al., 2023), we use a subset of it which contains 10000 samples in 50 classes as one of OOD datasets.

**Places.** Places (Zhou et al., 2017) is also a large scene dataset. Following previous methods (Huang & Li, 2021; Miyai et al., 2023a; Wang et al., 2023), we use a subset of it which contains 10000 samples in 50 classes as one of OOD datasets.

**Texture.** Texture (Cimpoi et al., 2014) is a open-world texture dataset. Following previous methods (Huang & Li, 2021; Miyai et al., 2023a; Wang et al., 2023), we use a subset of it which contains 5640 samples in 47 classes as one of OOD datasets.

## A.4 BASELINE AND IMPLEMENTATION DETAILS

### A.4.1 BASELINE DETAILS

In this section, we present the details of FS-OOD detection methods.

**CoOp.** CoOp (Zhou et al., 2022c) is a commonly used framework for prompt learning. Specifically, CoOp replaces a fixed textual prompt context with a set of learnable context vectors. During the fine-tuning process, only this small set of parameters needs to be trained and the encoder is frozen to adapt well to the downstream task. As a result, CoOp achieves efficient domain-adaptive migration.

**LoCoOp.** LoCoOp (Miyai et al., 2023a) is a classical CLIP-based FS-OOD detection method which utilizes local regularization. Specifically, LoCoOp performs OOD regularization by eliminating the influence of ID-irrelevant region information on text prompt during the training process. As a result, LoCoOp efficiently implements FS-OOD detection.

**SCT.** SCT (Yu et al., 2024) utilizes global self-calibrated tuning to improve FS-OOD detection performance without introducing additional parameters. Specifically, SCT utilizes the predicted probability for the ground-truth class to flexibly balance the importance of two tasks of LoCoOp. As a result, it reduces the impact of erroneous background information on text prompt, thus improving FS-OOD detection performance.

**Local-Prompt.** Local-Prompt (Zeng et al., 2025) is a negative prompt method designed for FS-OOD detection. Specifically, it leverages local outlier knowledge through negative enhancement guided by global prompt. In addition, it also utilizes region regularization enhanced by local prompt, effectively capturing local information. As a result, Local-Prompt demonstrates outstanding OOD detection performance and exhibits strong scalability.

Table 7: ImageNet-10 specific details

| n01530575 | Brambling | n01641577 | American bullfrog | n02107574 | Greater Swiss Mountain Dog | n02123597 | Siamese cat | n02389026 | Common sorrel horse |
| n02422699 | Impala (antelope) | n03095699 | Container ship | n03417042 | Garbage truck | n04285008 | Sports car | n04552348 | Military aircraft |

### A.4.2 IMPLEMENTATION DETAILS

**Experimental Environment.** All experiments are conducted with multiple runs on several NVIDIA A100 and NVIDIA GeForce RTX 4090 GPUs with Python 3.8.20, PyTorch 2.4.1, and CUDA 11.8.

**Training Details.** We adopt $\alpha = 1$ under all few-shot setting. The length of background prompt $L = 64$. The number $q$ of largest elements selected in $S_{\text{R-MCM}}$ is 10.

## A.5 ADDITIONAL EXPERIMENTAL RESULTS

Table 8: Experiments on ImageNet-1K as the ID dataset with 2-shot few-shot tuning results.

| Method | iNaturalist | | SUN | | Places | | Texture | | Average | |
|--------|-------------|---|-----|---|--------|---|---------|---|---------|---|
| | FPR95↓ | AUROC↑ | FPR95↓ | AUROC↑ | FPR95↓ | AUROC↑ | FPR95↓ | AUROC↑ | FPR95↓ | AUROC↑ |
| | | | | | *2-shot* | | | | | |
| $\text{CoOp}_{MCM}$ | $39.73^{\pm4.42}$ | $92.04^{\pm0.81}$ | $41.83^{\pm1.71}$ | $91.28^{\pm0.09}$ | $46.80^{\pm1.86}$ | $88.83^{\pm0.19}$ | $45.79^{\pm2.68}$ | $89.12^{\pm0.55}$ | $43.54^{\pm2.59}$ | $90.32^{\pm0.30}$ |
| $\text{CoOp}_{GL}$ | $19.15^{\pm3.36}$ | $95.70^{\pm0.85}$ | $31.72^{\pm3.18}$ | $92.02^{\pm0.51}$ | $39.31^{\pm3.48}$ | $89.19^{\pm0.76}$ | $48.54^{\pm1.61}$ | $85.87^{\pm1.01}$ | $34.68^{\pm2.34}$ | $90.70^{\pm0.26}$ |
| LoCoOp | $19.09^{\pm1.88}$ | $96.20^{\pm0.29}$ | $24.96^{\pm0.91}$ | $94.93^{\pm0.12}$ | $33.35^{\pm2.22}$ | $91.92^{\pm0.34}$ | $50.56^{\pm3.18}$ | $87.29^{\pm1.64}$ | $31.99^{\pm0.73}$ | $92.58^{\pm0.33}$ |
| Local-Prompt | $17.63^{\pm3.55}$ | $96.38^{\pm0.51}$ | $25.95^{\pm2.09}$ | $94.62^{\pm0.15}$ | $34.57^{\pm2.20}$ | $91.79^{\pm0.30}$ | $\mathbf{44.99}^{\pm3.31}$ | $\mathbf{89.52}^{\pm0.59}$ | $30.79^{\pm2.48}$ | $\mathbf{93.08}^{\pm0.26}$ |
| SCT | $15.58^{\pm2.41}$ | $96.66^{\pm0.45}$ | $25.48^{\pm0.67}$ | $94.11^{\pm0.04}$ | $34.11^{\pm1.36}$ | $91.31^{\pm0.11}$ | $48.09^{\pm3.06}$ | $86.81^{\pm1.12}$ | $30.82^{\pm1.80}$ | $92.22^{\pm0.38}$ |
| Mambo | $\mathbf{10.94}^{\pm1.46}$ | $\mathbf{97.61}^{\pm0.26}$ | $\mathbf{22.14}^{\pm0.64}$ | $94.80^{\pm0.47}$ | $\mathbf{30.51}^{\pm2.15}$ | $\mathbf{92.18}^{\pm0.72}$ | $46.40^{\pm1.49}$ | $87.68^{\pm0.91}$ | $\mathbf{27.50}^{\pm0.87}$ | $93.06^{\pm0.38}$ |

Table 9: Experiments on OpenOOD benchmark.

| Method | Near-OOD | | OOD | |
|--------|----------|---|-----|---|
| | FPR95↓ | AUROC↑ | FPR95↓ | AUROC↑ |
| LoCoOp | $91.18^{\pm0.45}$ | $49.27^{\pm0.38}$ | $58.58^{\pm0.94}$ | $78.67^{\pm0.35}$ |
| SCT | $90.58^{\pm0.51}$ | $51.05^{\pm0.63}$ | $58.49^{\pm0.53}$ | $78.75^{\pm0.56}$ |
| Local-Prompt | $88.76^{\pm0.46}$ | $50.18^{\pm0.25}$ | $\mathbf{55.24}^{\pm0.49}$ | $78.73^{\pm0.25}$ |
| Mambo | $\mathbf{87.97}^{\pm0.56}$ | $\mathbf{52.39}^{\pm0.16}$ | $56.53^{\pm1.81}$ | $\mathbf{79.68}^{\pm0.28}$ |

Table 10: Experiments on near OOD detection tasks with 4-shot few-shot tuning results.

| ID Dataset | OOD Dataset | Method | FPR95↓ | AUROC↑ |
|------------|-------------|--------|--------|--------|
| ImageNet-10 | ImageNet-20 | LoCoOp | $15.67^{\pm3.02}$ | $96.23^{\pm1.08}$ |
| | | Local-Prompt | $14.77^{\pm3.43}$ | $96.13^{\pm1.19}$ |
| | | SCT | $15.27^{\pm6.35}$ | $96.49^{\pm1.67}$ |
| | | Mambo | $\mathbf{13.43}^{\pm9.78}$ | $\mathbf{96.55}^{\pm2.07}$ |
| ImageNet-20 | ImageNet-10 | LoCoOp | $5.73^{\pm2.19}$ | $98.74^{\pm0.32}$ |
| | | Local-Prompt | $5.87^{\pm1.72}$ | $98.71^{\pm0.24}$ |
| | | SCT | $6.00^{\pm2.00}$ | $98.72^{\pm0.14}$ |
| | | Mambo | $\mathbf{4.60}^{\pm2.42}$ | $\mathbf{98.87}^{\pm0.38}$ |

## A.6 ADDITIONAL ABLATION ANALYSIS

**Computational Cost versus Detection Performance.** We report training time and memory consumption of our method compared with other baseline methods in Table 12. The evaluation is performed with a batch size as 32 on 1-shot. Experimental results demonstrate that our method introduces only few additional computational resources to obtain a significant improvement in FS-OOD detection performance compared with the baseline method SCT (Yu et al., 2024). Notably, our method achieves better performance on 1-shot while using just one-tenth of training time and less GPU memory than Local-Prompt. As a result, it demonstrates the efficiency of our method.

**OOD score strategy.** We investigate the differences in FS-OOD detection performance at different OOD score strategies. The experimental results reported in Figure 3 demonstrate that our method using R-MCM (Zeng et al., 2025) achieves better performance than MCM (Ming et al., 2022a) and GL-MCM (Miyai et al., 2025). Specifically, our method uses background prompt to guide FG-BG decomposition in local image features. As a result, background prompt also contains a portion of OOD potential local semantic information. Therefore, our method has a good compatibility with OOD score strategies related to local semantic information. As a result, it demonstrates the scalability and superiority of our method.

**Verification of the assumption made in patch SCT.** To further validate the hypothesis proposed in Patch SCT, we designed two sets of experiments. First, we compared the background region visualizations before and after suppressing high-accuracy samples. The results, which are shown in Figure 4, indicate that suppressing high-accuracy samples leads to more accurate identification of background regions. Second, we modified the implementation strategy of Patch SCT to apply adjustments only to low-accuracy samples and compared its detection performance with the original strategy on the ImageNet-1K benchmark. The results are shown in Table 13. Suppressing features extracted from high-accuracy samples effectively removes more ID-irrelevant features, thereby

Table 11: Trainable parameter of different methods on the ImageNet-1K OOD benchmark.

| Method | Trainable parameters (M) | FPR95↓ | AUROC↑ |
|---|---|---|---|
| LoCoOp | 0.0082 | $29.01^{\pm1.48}$ | $93.26^{\pm0.27}$ |
| SCT | 0.0082 | $27.78^{\pm1.56}$ | $92.82^{\pm0.48}$ |
| Local-Prompt | 9.8300 | $\mathbf{24.88}^{\pm0.24}$ | $\mathbf{94.40}^{\pm0.01}$ |
| Mambo ($N=8, L=8$) | 0.0082 | $26.46^{\pm0.48}$ | $93.51^{\pm0.11}$ |
| Mambo | 0.0410 | $25.31^{\pm0.30}$ | $93.68^{\pm0.20}$ |

Table 12: Time and memory cost of different methods on the ImageNet-1K OOD benchmark.

| Method | Time for one epoch (s) | GPU Memory (MiB) | FPR95↓ | AUROC↑ |
|---|---|---|---|---|
| LoCoOp | 21.63 | 19236 | $32.27^{\pm2.50}$ | $92.30^{\pm0.65}$ |
| SCT | 21.63 | 19236 | $33.56^{\pm4.25}$ | $91.46^{\pm1.20}$ |
| Local-Prompt | 270.9 | 24274 | $34.05^{\pm2.75}$ | $\mathbf{92.69}^{\pm0.41}$ |
| Mambo | 25.57 | 19252 | $\mathbf{30.38}^{\pm0.48}$ | $92.24^{\pm0.36}$ |

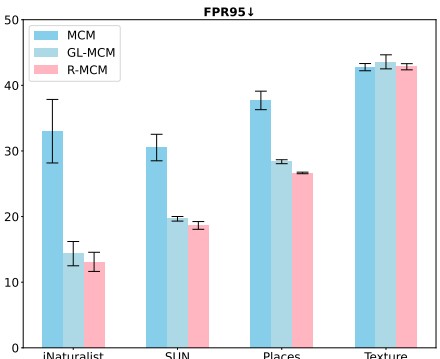
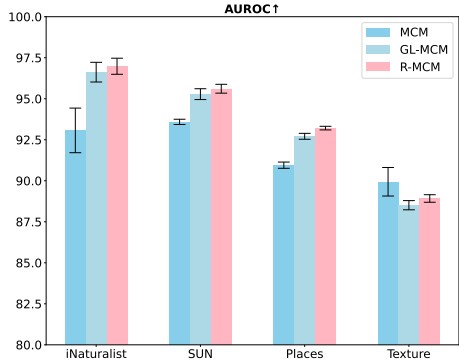

Figure 3: Ablation study of different OOD score strategies.

improving OOD detection performance. These two experiments together validate the correctness of the hypothesis proposed in Line 286.

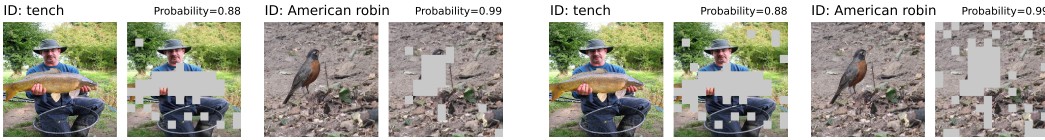

Figure 4: Visualization of extracted background patches: our method (left) and do not suppress high-accuracy samples (right).

## A.7 MORE VISUALIZATION

**Visualization of background prompt.** We report the visualization results of background prompt in Figure 5. The visualization results demonstrate that background prompt effectively attends to background regions other than the subject of ID features. For instance, background prompt successfully focuses on grass and branches in the hen class sample, which demonstrates that we improve FS-OOD detection performance by effectively utilizing the similarity between local image features and class text features through local similarity refinement.

**Visualization of distribution map.** We visualize the distribution map of four OOD datasets using ImageNet-1K as the ID dataset in Figure 6. The visualization results demonstrate that our method has a stronger discriminative ability for ID and OOD samples compared with the baseline method SCT (Yu et al., 2024). Specifically, the sample density of our method is more concentrated and the gap between ID and OOD scores is larger. This demonstrates that our method leveraging local similarity refinement and patch self-calibrated tuning to achieve superior background extraction, leading to more distinct differences between ID and OOD features.

**Visualization of no refinement.** We calculated the local similarity of images before and after the refinement operation and visualized the results. The results are shown in Figure 7 indicate that after applying the refinement operation, the blurry boundaries between background and foreground become clearer, and the impact of noise on background region selection is alleviated. For example, In a sample labeled as tench, after refinement, the area where the hand touches the fish is accurately identified as the background region, and noise from the trees in the background is greatly reduced. More visualization results will be added to the manuscript in subsequent revisions.

**Examples of failure.** Our method exhibits suboptimal performance when faced with some challenging samples, such as those where the target features are highly similar to the background regions or

Table 13: Experiments of verification on ImageNet-1K

| Method | iNaturalist | | SUN | | Places | | Texture | | Average | |
|---|---|---|---|---|---|---|---|---|---|---|
| | FPR95↓ | AUROC↑ | FPR95↓ | AUROC↑ | FPR95↓ | AUROC↑ | FPR95↓ | AUROC↑ | FPR95↓ | AUROC↑ |
| Mambo (w/o suppression) | $13.70^{\pm0.69}$ | $96.87^{\pm0.19}$ | $19.32^{\pm1.20}$ | $95.49^{\pm0.34}$ | $28.26^{\pm0.82}$ | $92.68^{\pm0.21}$ | $43.06^{\pm2.20}$ | $88.87^{\pm0.66}$ | $26.08^{\pm1.23}$ | $93.48^{\pm0.36}$ |
| Mambo | $\mathbf{13.10}^{\pm1.46}$ | $\mathbf{96.98}^{\pm0.49}$ | $\mathbf{18.65}^{\pm0.59}$ | $\mathbf{95.61}^{\pm0.27}$ | $\mathbf{26.65}^{\pm0.12}$ | $\mathbf{93.21}^{\pm0.11}$ | $\mathbf{42.83}^{\pm0.47}$ | $\mathbf{88.92}^{\pm0.23}$ | $\mathbf{25.31}^{\pm0.30}$ | $\mathbf{93.68}^{\pm0.20}$ |

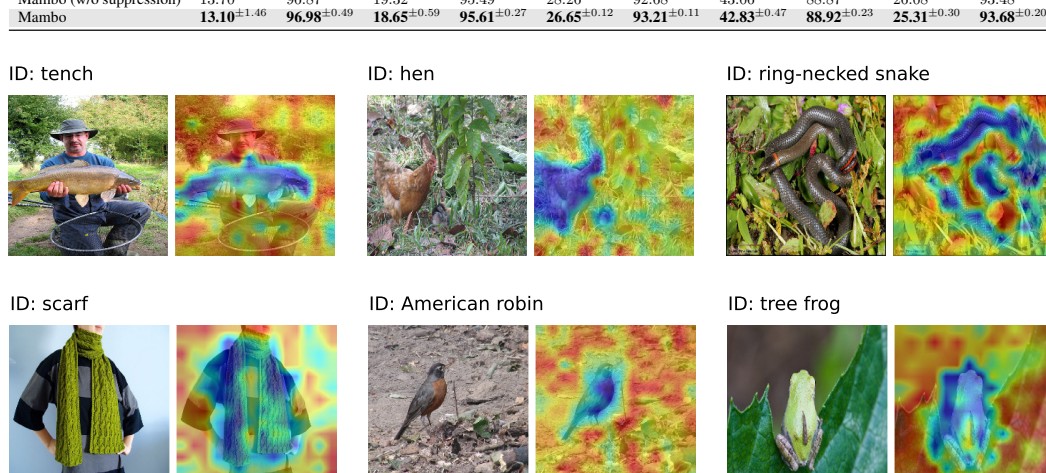

Figure 5: Visualization of background prompt.

lack distinctive characteristics. In such cases (*e.g.*, the examples are shown in Figure 8), background prompt often mistakenly identifies OOD features. Developing ways to differentiate between various types of features to enhance the model's performance on challenging data is a future research direction for us. Relevant failure visualizations will also be added to the manuscript in subsequent revisions.

**Difficult foreground samples.** Prior studies (Sun et al., 2024) have noted that FG-BG decomposition methods such as LoCoOp (Miyai et al., 2023a) may fail under challenging conditions *e.g.*, when an image contains multiple foreground objects or when the foreground is very small. Therefore, to evaluate the effectiveness of our proposed method on these challenging foreground samples, we visualize the local background similarity. As shown in Figure 10, the experimental results illustrate that even for these difficult cases, the local background similarity after the local similarity refinement still focuses on background regions while assigning low attention to different foreground objects. For example, in the tench class, where multiple fish overlap, the local background similarity is still able to correctly identify all foreground objects. In the nail class, although the snail occupies only a very small area, the local background similarity does not mistakenly classify additional background regions as foreground. Hence, the visualization results demonstrate both the robustness and the effectiveness of our proposed method.

**More visualization of background concept.** We further validate the learned background concepts through quantitative and more visualization. Specifically, we computed the cosine similarity between background text feature and class text features and used it to obtain the probability of the background feature for each class, visualized in Figure 11. The results show generally low and evenly distributed similarity, indicating that the learned background concepts are consistent and class-agnostic across categories. Additionally, we analyzed the affinity of ID and OOD samples to both class and background prompts. For each sample, the maximum similarity to class text features was taken as the class prompt affinity, and the similarity to background text feature as the background prompt affinity, shown in Figure 12. Results demonstrate that OOD samples generally have lower affinity to class prompts, while both ID and OOD samples show similar affinity to the background prompt. This confirms that the shared background prompt effectively captures class-agnostic background semantics without interfering with foreground classification boundaries. Slightly higher background affinity for OOD samples is likely due to their predominantly background content. These findings suggest that jointly leveraging class and background prompts enhances foreground semantics while mitigating background interference, leading to robust FS-OOD detection.

**Validation of motivation.** To demonstrate the limitations of existing FG–BG decomposition methods, specifically their heavy reliance on local class similarity and their fixed background patch extraction strategy. First, we select a set of low-quality samples (*e.g.*, those containing very small foreground objects ). For the FG–BG decomposition baseline LoCoOp, we visualize its local class similarity to

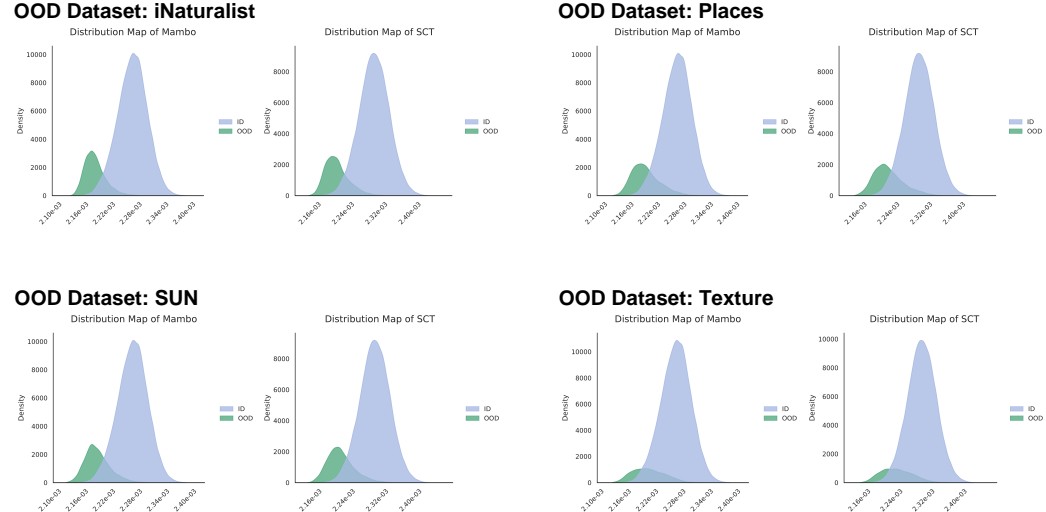

Figure 6: Distribution map of Mambo and SCT on four OOD datasets with ImageNet-1K as the ID dataset.

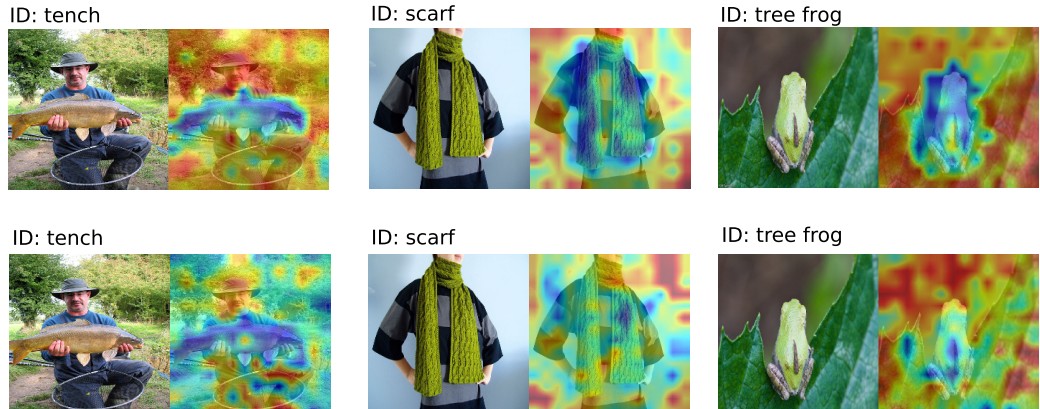

Figure 7: Visualization of the local similarity of images after the refinement operation (top) and before the refinement operation (bottom)

the ground-truth class. For Mambo, we visualize the local background similarity learned through background prompt. As shown in Figure 13, existing FG–BG methods tend to over-rely on local class similarity, which becomes unreliable when class prompts fail to capture meaningful foreground semantics. This leads to incorrect local similarity estimates and degraded OOD detection performance. In contrast, Mambo effectively identifies background regions through background prompt, allowing it to preserve foreground semantic information. For example, in the clownfish sample, where the foreground is extremely small and hard to detect, LoCoOp fails to locate the object and thus exhibits low similarity, whereas Mambo correctly responds to the background regions and preserves the foreground location. Second, to examine the issue of fixed background patch extraction strategies used in existing FG–BG methods, we compare the background patches selected by SCT and Mambo across samples with different classification accuracies. The visualization results are shown in Figure 14. We observe that methods with fixed background extraction fail to adapt to sample diversity and task requirements. For low-accuracy samples (*e.g.*, saharan horned viper), the model should retain more informative foreground regions to improve discriminative ability. However, SCT mistakenly treats part of the foreground as background and regularizes it, while Mambo preserves a larger set of informative patches. For high-accuracy samples (*e.g.*, sloth bear), where the model already has high confidence, SCT still retains a significant amount of background information, which may harm OOD detection. Mambo instead preserves only the core patches that provide richer OOD features and thereby improves robustness.

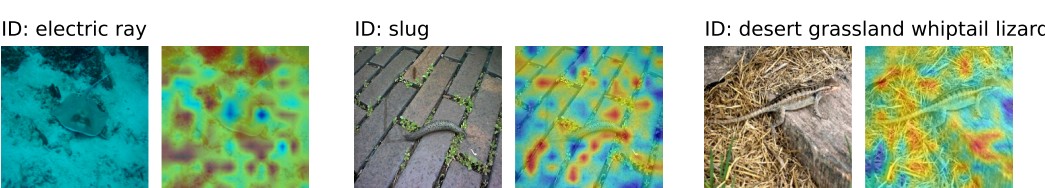

Figure 8: Visualization of the examples of failure

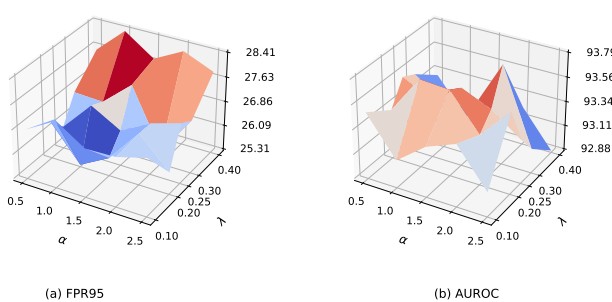

Figure 9: Visualization of hyperparameters $\lambda$ and $\alpha$ on ImageNet-1K OOD benchmark

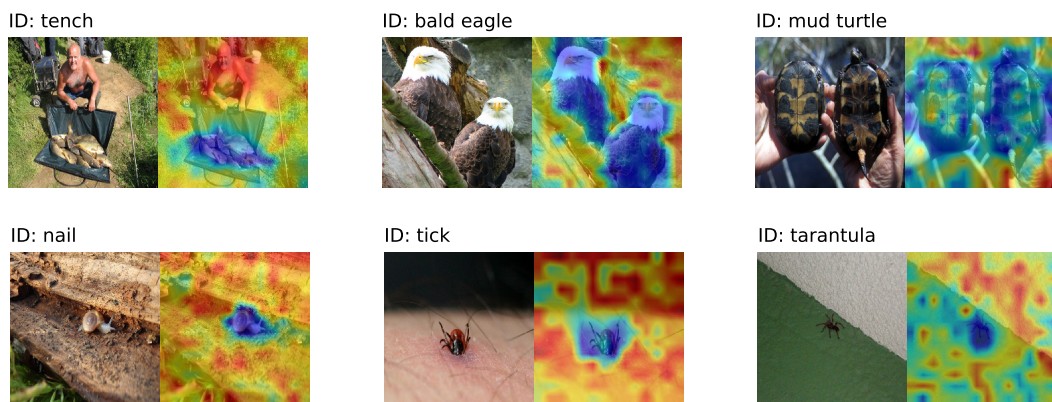

Figure 10: Visualization of the local similarity of images on examples with multiple foreground objects (top) and examples with very small foreground objects (bottom)

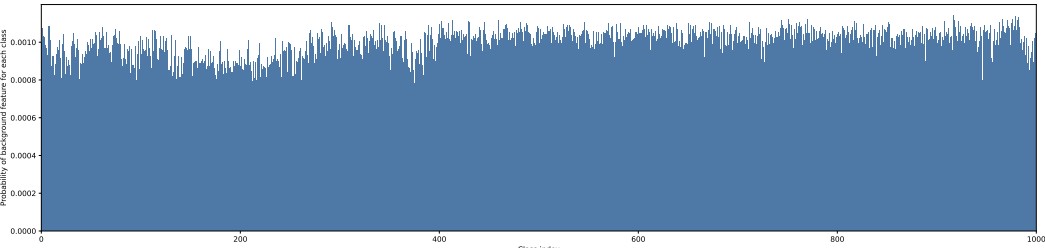

Figure 11: Probability of the background feature for each class

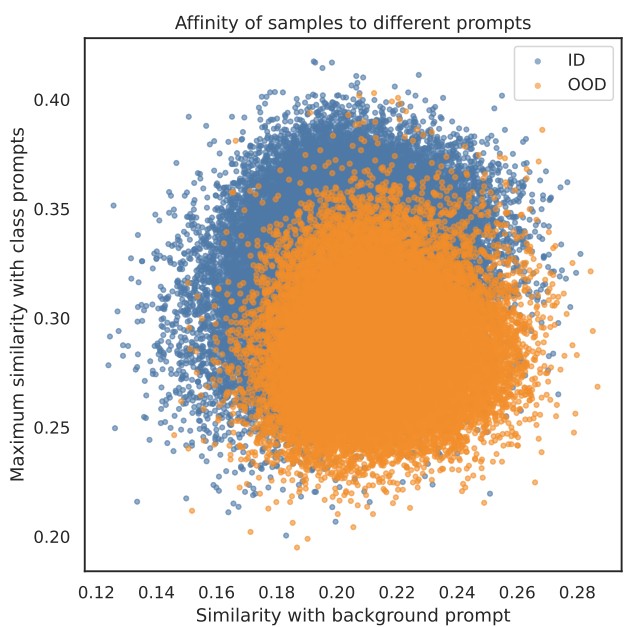

Figure 12: Visualization of affinity of samples to different prompts

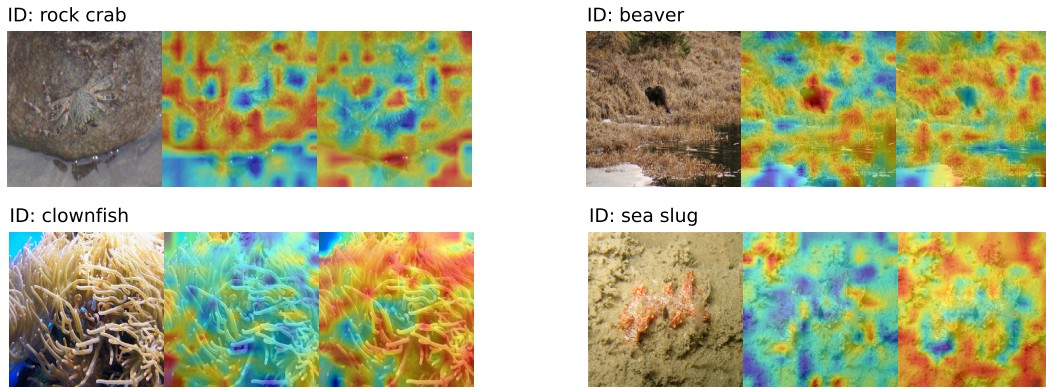

Figure 13: Visualization of the local similarity of images: LoCoOp (mid) and our method (right).

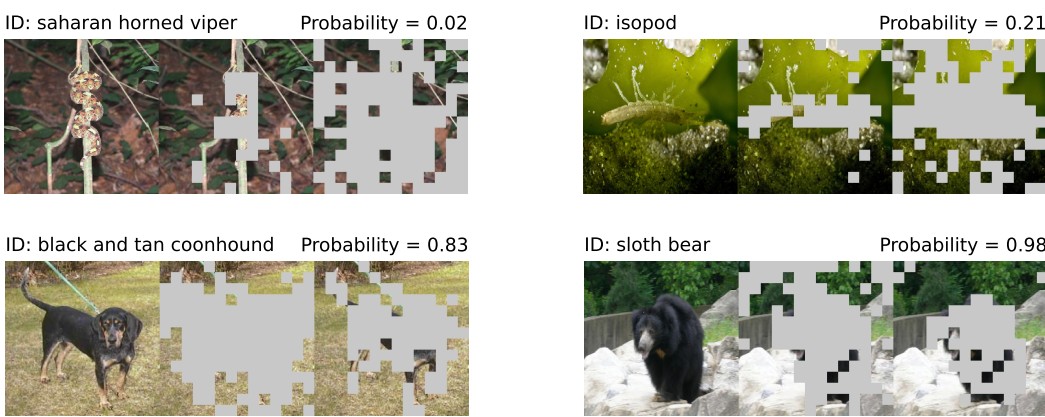

Figure 14: Visualization of extracted background patches: SCT (mid) and our method (right).

