# OpenReview forum: "Background Prompt for Few-Shot Out-of-Distribution Detection"
_ICLR.cc/2026/Conference — Submitted to ICLR 2026_

### Official Review · Reviewer_7DqH · 2025-10-28

**Soundness:** 2
**Presentation:** 3
**Contribution:** 2
**Rating:** 4
**Confidence:** 4

**Summary:**

This paper introduces Mambo, a framework for Few-Shot Out-of-Distribution (FS-OOD) detection. It addresses the limitations of existing FG-BG decomposition methods, such as over-reliance on local class similarity and fixed background patch extraction strategies. Mambo incorporates three key components:

1. A learnable prompt to extract background semantic information independently of class features.
2. A mechanism to refine background similarity using class similarity adaptively.
3. A dynamic strategy to extract a variable number of background patches tailored to each sample.

The authors validate Mambo on standard FS-OOD datasets and report that it outperforms state-of-the-art (SOTA) methods in terms of FPR95 and AUROC, while maintaining computational efficiency.

**Strengths:**

1. The paper effectively highlights key limitations in existing FG-BG decomposition methods, such as fixed patch selection strategies and over-reliance on local class similarity.
2. The empirical evaluation leverages multiple datasets (e.g., ImageNet-1K, iNaturalist, SUN, Places, Texture) and settings (e.g., OOD detection and near-OOD detection).
3. The ablation studies demonstrate the individual contributions of each proposed component.
4. Despite its improvements, Mambo remains computationally efficient, introducing minimal additional overhead compared to baseline methods like SCT and LoCoOp.
5. The paper provides detailed experimental settings and promises to release the source code, ensuring reproducibility.

**Weaknesses:**

1. This component is essentially a learnable prompt applied to the CLIP text encoder, which is a natural extension of existing prompt learning approaches (e.g., CoOp and Local-Prompt). The authors do not provide sufficient evidence that a learnable "background prompt" is superior to simpler alternatives, such as using predefined textual templates or leveraging existing negative prompts. For example, in Table 3, the ablation study shows that the baseline (without refinement or patch tuning) already achieves a relatively strong performance, with only modest improvements (e.g., FPR95 improves by ~2.2% on average) when adding the background prompt. This suggests that the background prompt alone does not bring transformative gains.

2. The refinement strategy combines local class similarity with background similarity in a weighted manner. While this is an effective mechanism, it lacks novelty, as it builds on existing weighting techniques commonly used in FS-OOD tasks (e.g., SCT's dynamic balancing). Moreover, The refinement weight $\delta_i$ in Eq (11) is heuristic and not well-justified.

3. The dynamic patch extraction strategy is conceptually similar to SCT's self-calibrated tuning. The primary difference is the introduction of a threshold parameter ($\alpha$) to adjust the number of extracted patches. While this adds flexibility, it is not a fundamentally new idea.

4. SCT also uses a self-calibrated tuning mechanism to dynamically balance tasks. The patch self-calibrated tuning in Mambo appears to be a direct extension of this idea, with the addition of a heuristic threshold (Eq (12)). The authors do not sufficiently highlight how their approach is distinct or superior.

5. Local-Prompt leverages negative prompts and local outlier knowledge for FS-OOD detection. The background prompt in Mambo serves a similar purpose but focuses on extracting background features instead of negatives. The novelty here is limited, as both approaches aim to augment prompt learning with auxiliary prompts.

6. Several components, such as the refinement weight ($\delta_i$) and the threshold parameter ($\alpha$), are heuristic and not rigorously justified. This raises concerns about the robustness of the method across different datasets and tasks. For example, the hyperparameter sensitivity analysis (Fig. 9) shows that small changes in $\alpha$ can lead to noticeable performance drops, indicating that the method may require careful tuning.

Overall, the paper tackles important challenges in FS-OOD detection and shows good empirical results, but the contributions feel incremental, and the added complexity isn't fully justified. A broader evaluation and clearer differentiation from prior work would better highlight its significance.

**Questions:**

See Weaknesses

---

> ### Author Response · Authors · 2025-11-20
> **Official Comment by Authors**
>
> Thank you for taking the time to review our paper and for your constructive feedback. Below is our detailed response to your comments. All modifications will be reflected in the revised manuscript.
>
> >**Q1: This component is essentially a learnable prompt applied to the CLIP text encoder, which is a natural extension of existing prompt learning approaches (e.g., CoOp and Local-Prompt). The authors do not provide sufficient evidence that a learnable "background prompt" is superior to simpler alternatives, such as using predefined textual templates or leveraging existing negative prompts. For example, in Table 3, the ablation study shows that the baseline (without refinement or patch tuning) already achieves a relatively strong performance, with only modest improvements (e.g., FPR95 improves by ~2.2% on average) when adding the background prompt. This suggests that the background prompt alone does not bring transformative gains.**
>
> **A1:** We **clarify its contribution from three perspectives**:
>
> **1. The background prompt provide a new type of semantics that previous methods cannot model: a conceptual representation of the background**. Existing methods mainly model semantic information in three ways:
>
> **(1) Using predefined textual templates**, which can explicitly encode semantics, but constructing background semantics in this way requires **additional external information** (e.g., using large language models to generate background concepts), thereby reducing the generalizability of the model.
>
> **(2) Using CoOp learnable contexts to automatically learn background semantics**, but CoOp must embed class labels into the textual prompt, which may cause the learned background information to **be entangled with specific classes**.
>
> **(3) Using negative prompts as in Local-Prompt**, which can learn class-agnostic semantics, but such methods often require **a large number of negative prompts**, resulting in **high computational cost**. Moreover, the negative prompts used in Local-Prompt mainly aim to learn hard negative samples, whose features contain both background and non-salient foreground objects, **making it difficult to cleanly separate background features**.
>
> As shown in the visualization in Figure. 5, the background prompt learns response regions that highly match the true background semantics. This represents a capability-level addition, rather than only a performance improvement.
>
> **2.Performance analysis of the background prompt.** As shown in the Table 3, our baseline **includes the background prompt** to provide additional background semantics; the subsequent ~2.2% improvement comes from **the newly added local similarity refinement and patch self-calibrated tuning modules**. **It is important to emphasize that a **2.2%** improvement is highly significant in OOD tasks**, for example, **the improvement of LoCoOp over CoOp is around 2%**. In addition, the background prompt provides the necessary semantic decomposition ability for the model, which is the premise that allows the later modules to function, so its contribution is reflected not only in numerical gains but also in its role as **a fundamental component** of the overall method.
>
> **3.From the experiments**, we can observe that **the background prompt significantly improves the model’s robustness under extreme scenarios** (e.g., multiple and small  foregrounds). In the additional visualization (**Figure. 13**), we clearly show that in cases where the foreground is extremely weak or the background contains strong interference, existing FG-BG decomposition methods often **fail to discriminate foreground objects**, whereas the background prompt can **correctly focus on background regions**, enabling the model to preserve foreground information and thus avoid a substantial drop in OOD performance.

---

> ### Author Response · Authors · 2025-11-20
>
> >**Q2: The refinement strategy combines local class similarity with background similarity in a weighted manner. While this is an effective mechanism, it lacks novelty, as it builds on existing weighting techniques commonly used in FS-OOD tasks (e.g., SCT's dynamic balancing). Moreover, The refinement weight  in Eq (11) is heuristic and not well-justified.**
>
> **A2:** We would like to emphasize **both the novelty of our local similarity refinement and the rationale behind our weighting design**. Although both our method and SCT use accuracy for adaptive weighting, this mechanism is a widely adopted and is not intended to be a core contribution by itself. **Our contribution lies in the fact that the semantics and motivation behind the weight are fundamentally different**.
>
> First, SCT uses accuracy to decide which **training objective** the model should prioritize, focusing more on classification when accuracy is low, and more on OOD detection when accuracy is high. In contrast, we **use accuracy as an indicator of sample quality**: higher accuracy implies more reliable local class similarity, which should be more strongly incorporated when refining the local background similarity. Thus, **our weighting mechanism is not about making a trade-off between two tasks; instead, it dynamically decides, at the local feature level, whether class semantics or background semantics should be more trusted**. The **semantic meaning** and **operational role of the weight** are therefore **entirely different** from those in SCT.
>
> Second, **we further clarify the formulation and interpretation of our refinement weight to justify its design**. Specifically, our goal is to refine local background similarity using class semantics, without letting class similarity dominate the refinement. Thus, we use class similarity only as a weighting signal and apply two essential adjustments: **(1) Because local class and background similarity has different scales, we normalize the weights to prevent local class similarity from overwhelming the refinement; (2) Since the two signals encode opposite semantics, we use a negative weighting scheme patches with lower local class similarity (more likely background) receive higher refinement weights**. This ensures that the refinement behaves semantically correctly while remaining background-driven.
>
> >**Q3, Q4:  The novelty of patch self-calibrated tuning.**
>
> **A3, A4:** We would like to highlight **the differences between our patch self-calibrated tuning and SCT**. SCT employs self-calibrated tuning to adjust the training objective, thereby **mitigating performance degradation on high-uncertainty samples**. However, SCT still **focuses on adjustments at the task and global levels**. As a result, **SCT cannot avoid extracting incorrect background patches during background extraction**; it can only **alleviate the impact** of these erroneous patches through objective adjustment.
> In contrast, our patch self-calibrated tuning module operates directly at **the patch level**. By adopting an adaptive background extraction strategy, **it effectively reduces the extraction of incorrect background patches and fundamentally decreases the likelihood of such errors**. Moreover, as shown in the newly added visualization results (**Figure. 14**), our method provides **a more intuitive view of how task objectives evolve and enables more precise objective adjustment**. This not only demonstrates the effectiveness of our patch self-calibrated tuning but also further highlights the **unique advantages of our proposed method**.

---

> ### Author Response · Authors · 2025-11-20
>
> >**Q5: Local-Prompt leverages negative prompts and local outlier knowledge for FS-OOD detection. The background prompt in Mambo serves a similar purpose but focuses on extracting background features instead of negatives. The novelty here is limited, as both approaches aim to augment prompt learning with auxiliary prompts.**
>
> **A5:** We would like to highlight **the differences between Mambo and Local-Prompt**. Although Local-Prompt also introduces auxiliary prompts, **the design logic of its negative prompts fundamentally differs from our proposed background prompt in terms of implementation, semantic representation, and modeling capability**.
> In terms of implementation, Local-Prompt relies on **extensive downsampling and a large number of prompts** to adequately learn hard negative samples. As shown in Table 11, Table 12, and the table below, we compare the parameter count and computational cost of Mambo with other methods.
>
> | Method| Trainable Parameters (M) | FPR95 ↓| AUROC ↑|
> |-|-|-|-|
> | LoCoOp| 0.0082| 29.01 ± 1.48 | 93.26 ± 0.27|
> | SCT| 0.0082 | 27.78 ± 1.56 | 92.82 ± 0.48|
> | Local-Prompt | 9.8300| 24.88 ± 0.24   | 94.40 ± 0.01|
> | Mambo (N = 8, L = 8) | 0.0082| 26.46 ± 0.48| 93.51 ± 0.11|
> | Mambo | 0.0410 | 25.31 ± 0.30| 93.68 ± 0.20|
>
> | Method        | Time for One Epoch (s) | GPU Memory (MiB) | FPR95 ↓        | AUROC ↑       |
> |---------------|-------------------------|-------------------|----------------|----------------|
> | LoCoOp        | 21.63   | 19236             | 32.27 ± 2.50   | 92.30 ± 0.65   |
> | SCT           | 21.63      | 19236             | 33.56 ± 4.25   | 91.46 ± 1.20   |
> | Local-Prompt  | 270.9    | 24274     | 34.05 ± 2.75   | 92.69 ± 0.41   |
> | Mambo         | 25.57                   | 19252             | 30.38 ± 0.48   | 92.24 ± 0.36   |
>
> The results indicate that Local-Prompt has a **substantial computational and memory overhead**, which is a major limitation in practical applications. In contrast, our background prompt is a **lightweight, single learnable vector**, which not only uses far fewer parameters than Local-Prompt but also significantly reduces training and inference costs.
>
> More importantly, the negative prompts in Local-Prompt **cannot provide clean and stable background semantics**. They learn hard negative features that include **not only background information but also a significant amount of non-prominent ID foreground features**, resulting in highly mixed and uncontrollable semantics. Consequently, extracting a pure background concept from negative prompts is difficult, which limits their expressiveness in FG–BG decomposition scenarios. In contrast, our background prompt is explicitly designed to learn background as an independent semantic concept, and its activation regions in visualizations align closely with the true background.
>
> In summary, although both methods can be seen as extended prompt mechanisms in form, they **fundamentally differ** in terms of target semantics, modeling capability, computational overhead. Therefore, the background prompt should not be regarded as a simple variant or minor adaptation of Local-Prompt.
>
> >**Q6: Several components, such as the refinement weight ($\Delta_i$) and the threshold parameter ($\alpha$), are heuristic and not rigorously justified. This raises concerns about the robustness of the method across different datasets and tasks. For example, the hyperparameter sensitivity analysis (Fig. 9) shows that small changes in  can lead to noticeable performance drops, indicating that the method may require careful tuning.**
>
> **A6:** We would like to clarify the hyperparameter settings of Mambo to address the reviewer’s concerns regarding robustness. First, the design of the refinement weight $\Delta_i$ has been detailed in Q2. Regarding the threshold parameter $\alpha$, it is designed to control the degree of adaptiveness: it is bidirectional, where a larger $\alpha$ corresponds to stronger adaptiveness, extracting more patches for low-accuracy samples and removing more background regions for high-accuracy samples. Since $\alpha$ directly controls the multiple of the standard deviation, large adjustments naturally have a significant impact on OOD detection performance. However, this does not compromise the robustness of Mambo. **In fact, across all experimental settings, except for the number of epochs, we used the same hyperparameters**, and the results show that Mambo consistently **achieves the best performance even without extensive hyperparameter tuning**, demonstrating its **robustness across different dataset settings**. Furthermore, as shown in the newly added **Table 4**, we compare Mambo’s performance across different CLIP backbones. The results show that Mambo **outperforms the best FG–BG decomposition methods regardless of the CLIP backbone**. Notably, we used the same hyperparameters for all architectures, which further **confirms the robustness of Mambo’s hyperparameters across different VLM architectures**.

---

> ### Author Response · Authors · 2025-11-27
> **Looking forward to your feedback**
>
> Dear reviewer 7DqH:
>
> We respectfully appreciate again for your insightful and thoughtful comments! As the suggestions from the reviewer, we give thorough explanation and update the manuscripts accordingly.
>
> As it is approaching the end of the discussion period (December 3 at 11:59pm AoE) and we do not receive feedback, we sincerely hope you could look through our response and have a further comment at your convenience if you have any questions about the paper. We will do our best to address the issues of the reviewer.
>
> Best wishes,
>
> Submission 12004 Authors.

---

### Official Review · Reviewer_vd43 · 2025-10-31

**Soundness:** 3
**Presentation:** 3
**Contribution:** 3
**Rating:** 8
**Confidence:** 3

**Summary:**

The authors propose a foregroudn-background decomposition framework for few-shot out-of-distribution  detection. The aim is to address the challenge of low robustness due to over-reliance on the local class similarity and a fixed background patch extraction.
 .

**Strengths:**

- The paper is well organized and written.
- The proposed method named Mambo is detailed and reproducible.
- The contribution addresses  a key challenge in the field.
- Experiments are well conducted showing the superiority of Mambo compared  to three previous methods.

**Weaknesses:**

None

**Questions:**

Please remind the citation [] after each acronym of the compared methods in Table 2.

---

> ### Author Response · Authors · 2025-11-20
> **Official Comment by Authors**
>
> Thank you for taking the time to review our paper and for your constructive feedback. Below is our detailed response to your comments. All modifications will be reflected in the revised manuscript.
>
> >**Q1: Please remind the citation [] after each acronym of the compared methods in Table 2.**
>
> **A1:** We thank the reviewer for the positive feedback and the valuable comments. In the revised manuscript, we have **added citations for each comparative method in Table 2**. If there are any further questions or concerns regarding our work, we would be happy to discuss them.

---

### Official Review · Reviewer_VmDp · 2025-10-31

**Soundness:** 2
**Presentation:** 2
**Contribution:** 2
**Rating:** 2
**Confidence:** 5

**Summary:**

This paper investigates the problem of erroneous OOD feature extraction in existing few-shot OOD detection methods and argues that over-reliance on the local class similarity and a fixed background patch extraction strategy are the two main reasons. It proposes Mambo, a new foreground-background decomposition framework which consists of a learnable background prompt refined by local class probabilities and patch self-calibrated tuning to adaptively extract background information. Experiments on ImageNet OOD benchmarks and OpenOOD benchmarks are conducted to verify its effectiveness.

**Strengths:**

1. The problem of erroneous OOD feature extraction in existing few-shot OOD detection methods is interesting.
2. Experiments on large-scale ImageNet OOD benchmarks and OpenOOD benchmarks are conducted to evaluate the proposed method.

**Weaknesses:**

1. **The presentation of the proposed method is incomplete and unclear**. First, the proposed background prompt is learnable but there is no mathematical formulation and explanation of the loss function to optimize the learnable prompt. Second, the meaning of $s_j^{class}$ in Eq.(11) is not clarified and there is no in-depth discussion about the design and implication of the refinement weight factor $\Delta_i$. It’s also recommended that the authors provide a clear algorithmic summary of the pipeline of **Mambo**.
2. **The OOD detection performance of the proposed method is not compelling**. For instance, on the ImageNet-1k OOD benchmark, **Mambo** only outperforms the best baseline, Local-Prompt, by less than 1 in terms of FPR95 and falls behind Local-Prompt in terms of AUROC under the 4-shot setting and falls behind Local-Prompt in terms of these two metrics under the 16-shot setting.
3. To **validate the motivation of this paper**, the reviewer suggests that the authors provide **sufficient illustrations or experimental results** to demonstrate the problem of over-reliance on the local class similarity and fixed background patch extraction strategies.
4. To **further verify the effectiveness** of the proposed method, the reviewer suggests that the authors add experiments on **different CLIP backbone structures**.

**Questions:**

Please see the weaknesses.

**Details Of Ethics Concerns:**

The reviewer doesn't notice any ethical issue with this paper.

---

> ### Author Response · Authors · 2025-11-20
> **Official Comment by Authors**
>
> Thank you for taking the time to review our paper and for your constructive feedback. Below is our detailed response to your comments. All modifications will be reflected in the revised manuscript.
>
> >**Q1: The presentation of the proposed method is incomplete and unclear. First, the proposed background prompt is learnable but there is no mathematical formulation and explanation of the loss function to optimize the learnable prompt. Second, the meaning of  in Eq.(11) is not clarified and there is no in-depth discussion about the design and implication of the refinement weight factor . It’s also recommended that the authors provide a clear algorithmic summary of the pipeline of Mambo.**
>
> **A1:** Thank you for the reviewer’s careful reading and constructive comments.
>
> **(1) Regarding the optimization of background prompt**, it is optimized jointly through the loss terms $L_{\text{CE}}$ and $L_{\text{OOD}}$. We have added more detailed explanations in the revised manuscript to clarify how the background prompt is learned.
>
> **(2) Regarding the refinement weight $s_j^{\text{class}}$**, it represents the local class similarity of the $j$-th patch.
>
> **(3) Regarding the design of  refinement weight factor $\Delta_i$**, its design goal is to use rich class-semantic information to refine the local background similarity. Therefore, the refinement should still be **primarily governed by the local background similarity rather than the local class similarity**. To achieve this, we **use the local class similarity as a weighting factor** in a controlled, weighted-refinement manner. However, this formulation presents two challenges: First, **the magnitudes of local class similarity and local background similarity differ significantly**. Direct weighting would cause the final result to be dominated by local class similarity. To prevent this, we apply a normalization step to constrain all weights within **a fixed range** and avoid such domination. Second, local class similarity and local background similarity **encode opposite semantics**. Using the raw class-similarity value as a weight would therefore lead to the opposite of the desired refinement effect. To address this, we use **a negative weighting strategy**: patches with lower local class similarity are more likely to belong to background regions, and thus should contribute more to the refinement. Accordingly, after normalization, we negate the class-similarity signal to obtain a semantically correct refinement weight.
> Combining these design motivations, the refinement factor represents the **class-semantic information** we choose to leverage for each local region, obtained by normalizing and negating the local class similarity.
>
> **(4)** We have also included a complete algorithmic summary in **Algorithm 1** to provide a clearer and more intuitive description of Mambo.

---

> ### Author Response · Authors · 2025-11-20
>
> >**Q2: The OOD detection performance of the proposed method is not compelling. For instance, on the ImageNet-1k OOD benchmark, Mambo only outperforms the best baseline, Local-Prompt, by less than 1 in terms of FPR95 and falls behind Local-Prompt in terms of AUROC under the 4-shot setting and falls behind Local-Prompt in terms of these two metrics under the 16-shot setting.**
>
> **A2:** We address the reviewer’s concerns as follows.
>
> **1. Performance comparison with Local-Prompt.** While Mambo achieves comparable results to Local-Prompt under the general ImageNet-1k OOD benchmark, it is important to highlight that in the **1-shot setting**, where **semantic information is extremely limited**. Mambo reduces FPR95 by **3.67%** compared to Local-Prompt. This demonstrates that Mambo is particularly effective in **low-data regimes**, which are **crucial for few-shot OOD detection**.
>
> **2. Fair comparison within the FG–BG decomposition paradigm.** **Mambo is fundamentally built upon a FG-BG decomposition framework**. Therefore, a more appropriate comparison should be made among methods within this paradigm. **Across all shot configurations**, Mambo consistently **outperforms existing FG–BG decomposition few-shot OOD detection methods**. For example, in the **16-shot setting**, compared with **the best FG–BG method SCT**: Mambo reduces FPR95 by **2.47%** and improves AUROC by **0.86%**. These results indicate that our learned background concepts and local similarity refinement significantly enhance FG–BG decomposition based OOD detection.
>
> **3. Efficiency advantages: parameter size and computational cost.** Beyond performance, we emphasize that **Mambo** is designed to be **a highly efficient method**. We provide two comparison tables in the paper.
>
> Firstly, the following table shows that Mambo uses only **1/200** of the trainable parameters of **Local-Prompt** while achieving competitive performance.
> | Method                 | Trainable Parameters (M) | FPR95 ↓        | AUROC ↑       |
> |------------------------|---------------------------|----------------|----------------|
> | LoCoOp                 | 0.0082                    | 29.01 ± 1.48   | 93.26 ± 0.27   |
> | SCT                    | 0.0082                    | 27.78 ± 1.56   | 92.82 ± 0.48   |
> | Local-Prompt           | 9.8300                    | 24.88 ± 0.24   | 94.40 ± 0.01   |
> | Mambo (N = 8, L = 8)   | 0.0082                    | 26.46 ± 0.48   | 93.51 ± 0.11   |
> | Mambo                  | 0.0410                    | 25.31 ± 0.30   | 93.68 ± 0.20   |
>
> Secondly, the following table further demonstrates that Mambo requires substantially fewer computational resources than Local-Prompt. Notably, the large parameter count and high computational overhead are inherent limitations of negative prompt–based methods such as Local-Prompt. Our results show that Mambo overcomes these limitations while maintaining strong OOD detection capability.
> | Method        | Time for One Epoch (s) | GPU Memory (MiB) | FPR95 ↓        | AUROC ↑       |
> |---------------|-------------------------|-------------------|----------------|----------------|
> | LoCoOp        | 21.63                   | 19236             | 32.27 ± 2.50   | 92.30 ± 0.65   |
> | SCT           | 21.63                   | 19236             | 33.56 ± 4.25   | 91.46 ± 1.20   |
> | Local-Prompt  | 270.9                   | 24274             | 34.05 ± 2.75   | 92.69 ± 0.41   |
> | Mambo         | 25.57                   | 19252             | 30.38 ± 0.48   | 92.24 ± 0.36   |
>
> In summary, **Mambo is both effective and efficient**: it shows stronger performance in **extremely low-shot regimes**, achieves SOTA results among FG–BG decomposition few-shot OOD detection methods, and requires dramatically **fewer parameters and computational resources** compared to baseline methods such as Local-Prompt. We believe these advantages highlight the practical value and conceptual contribution of Mambo.

---

> ### Author Response · Authors · 2025-11-20
>
> >**Q3: To validate the motivation of this paper, the reviewer suggests that the authors provide sufficient illustrations or experimental results to demonstrate the problem of over-reliance on the local class similarity and fixed background patch extraction strategies.**
>
> **A3:**  Thank you for the valuable suggestions. To demonstrate the limitations of existing FG–BG decomposition methods, specifically their heavy reliance on local class similarity and their fixed background patch extraction strategy, **we provide two additional visualization experiments to further validate the motivation of our work**.
>
> First, we select a set of **low-quality samples** (e.g., those containing very small foreground objects ). For the FG–BG decomposition baseline LoCoOp, **we visualize its local class similarity to the ground-truth class. For Mambo, we visualize the local background similarity learned through background prompt**. As shown in **Figure 13**, existing FG–BG methods tend to **over-rely on local class similarity**, which becomes unreliable when **class prompts fail to capture meaningful foreground semantics**. This leads to incorrect local similarity estimates and degraded OOD detection performance. **In contrast, Mambo effectively identifies background regions through background prompt, allowing it to preserve foreground semantic information.** For example, in the clownfish sample, where the foreground is extremely small and hard to detect, LoCoOp fails to locate the object and thus exhibits low similarity, whereas Mambo correctly responds to the background regions and preserves the foreground location.
>
> Second, **to examine the issue of fixed background patch extraction strategies used in existing FG–BG methods, we compare the background patches selected by SCT and Mambo across samples with different classification accuracies**. The visualization results are shown in **Figure 14**. We observe that **methods with fixed background extraction fail to adapt to sample diversity and task requirements**. For low-accuracy samples (e.g., saharan horned viper), the model should retain more informative foreground regions to improve discriminative ability. However, SCT mistakenly treats part of the foreground as background and regularizes it, while **Mambo preserves a larger set of informative patches**. For high-accuracy samples (e.g., sloth bear), where the model already has high confidence, SCT still retains a significant amount of background information, which may harm OOD detection. **Mambo instead preserves only the core patches that provide richer OOD features and thereby improves robustness**.
>
> Finally, to further verify this behavior, we include an additional ablation comparing Mambo with and without  suppression on high accuracy samples in the following table.
>
> | Method                   | iNaturalist FPR95↓ | iNaturalist AUROC↑ | SUN FPR95↓ | SUN AUROC↑ | Places FPR95↓ | Places AUROC↑ | Texture FPR95↓ | Texture AUROC↑ | Average FPR95↓ | Average AUROC↑ |
> |--------------------------|--------------------|---------------------|------------|-------------|----------------|----------------|-----------------|------------------|------------------|------------------|
> | Mambo (w/o suppression) | 13.70±0.69        | 96.87±0.19          | 19.32±1.20 | 95.49±0.34  | 28.26±0.82     | 92.68±0.21     | 43.06±2.20      | 88.87±0.66       | 26.08±1.23       | 93.48±0.36       |
> | Mambo                    | 13.10±1.46        | 96.98±0.49          | 18.65±0.59 | 95.61±0.27  | 26.65±0.12     | 93.21±0.11     | 42.83±0.47      | 88.92±0.23       | 25.31±0.30       | 93.68±0.20       |
>
>
> The results show that retaining more background patches for high accuracy samples helps improve OOD detection performance, which is consistent with our design motivation.

---

> ### Author Response · Authors · 2025-11-20
>
> >**Q4: To further verify the effectiveness of the proposed method, the reviewer suggests that the authors add experiments on different CLIP backbone structures.**
>
> **A4:** To validate the effectiveness of our method, we conducted experiments across **multiple CLIP backbone architectures** and compared our method with the best FG–BG decomposition method, SCT. The results are shown in the following table, demonstrate that our method consistently achieves superior performance across all architectures, indicating that Mambo outperforms the best FG–BG decomposition methods regardless of the CLIP backbone. Notably, we used the same hyperparameters for all architectures, which further demonstrates the robustness of Mambo’s hyperparameters across different VLM architectures.
>
> | Method | Architecture | iNaturalist |  | SUN |  | Places |  | Texture |  | Average |  |
> |--------|--------------|------------|----|-----|----|--------|----|--------|----|--------|----|
> |  |  | FPR95 | AUROC | FPR95 | AUROC | FPR95 | AUROC | FPR95 | AUROC | FPR95 | AUROC |
> | SCT    | ViT-B/32 | 28.46 ± 4.89 | 94.83 ± 0.71 | 35.18 ± 3.84 | 92.90 ± 0.54 | 41.25 ± 2.84 | 90.62 ± 0.44 | 48.15 ± 2.62 | 88.32 ± 0.76 | 38.26 ± 2.71 | 91.67 ± 0.34 |
> | Mambo  | ViT-B/32 | 26.75 ± 2.02 | 95.02 ± 0.38 | 33.80 ± 1.43 | 93.22 ± 0.25 | 40.45 ± 1.35 | 90.75 ± 0.26 | 46.73 ± 0.45 | 88.46 ± 0.07 | 36.93 ± 1.03 | 91.87 ± 0.15 |
> | SCT    | ViT-B/16 | 15.33 ± 2.24 | 96.77 ± 0.28 | 20.93 ± 1.51 | 95.10 ± 0.29 | 29.98 ± 1.15 | 92.19 ± 0.04 | 44.88 ± 1.70 | 88.20 ± 0.42 | 27.78 ± 1.56 | 92.82 ± 0.48 |
> | Mambo  | ViT-B/16 | 13.10 ± 1.46 | 96.98 ± 0.49 | 18.65 ± 0.59 | 95.61 ± 0.27 | 26.65 ± 0.12 | 93.21 ± 0.11 | 42.83 ± 0.47 | 88.92 ± 0.23 | 25.31 ± 0.30 | 93.68 ± 0.20 |
> | SCT    | ViT-L/14 | 31.95 ± 1.93 | 93.44 ± 0.46 | 30.71 ± 0.80 | 93.01 ± 0.62 | 35.34 ± 1.46 | 91.09 ± 0.50 | 49.62 ± 1.61 | 85.68 ± 0.37 | 36.91 ± 0.77 | 90.81 ± 0.27 |
> | Mambo  | ViT-L/14 | 28.21 ± 1.80 | 94.44 ± 0.39 | 29.88 ± 1.60 | 93.38 ± 0.23 | 35.29 ± 1.23 | 91.41 ± 0.07 | 47.68 ± 0.77 | 87.36 ± 0.39 | 35.27 ± 0.52 | 91.65 ± 0.15 |

---

> ### Author Response · Authors · 2025-11-27
> **Looking forward to your feedback**
>
> Dear reviewer VmDp:
>
> We respectfully appreciate again for your insightful and thoughtful comments! As the suggestions from the reviewer, we give thorough explanation and update the manuscripts accordingly.
>
> As it is approaching the end of the discussion period (December 3 at 11:59pm AoE) and we do not receive feedback, we sincerely hope you could look through our response and have a further comment at your convenience if you have any questions about the paper. We will do our best to address the issues of the reviewer.
>
> Best wishes,
>
> Submission 12004 Authors.

---

### Official Review · Reviewer_uEb4 · 2025-10-31

**Soundness:** 2
**Presentation:** 2
**Contribution:** 2
**Rating:** 4
**Confidence:** 4

**Summary:**

The paper focuses on FS-OOD detection with VLMs. The authors argue that existing FG-BG methods (e.g., LoCoOp, SCT) suffer from: (1) heavy dependence on potentially inaccurate local class similarityfor background identification, and (2) inflexible background extraction strategies (e.g., fixed top-K) that ignore sample diversity. To tackle these, Mambo proposes three core components. The method is evaluated on standard OOD and near-OOD tasks, outperforming existing zero-shot and few-shot baselines. Ablation studies and visualizations validate the design choices.

**Strengths:**

1. The paper accurately identifies the core weaknesses of FG-BG methods, providing a clear rationale for the proposed solutions.
2. The background prompt is a creative idea to model background semantics explicitly. The integration of refinement and self-calibration is logically structured and directly addresses the stated limitations.
3. Extensive experiments cover diverse datasets (e.g., ImageNet-1K, near-OOD benchmarks), few-shot settings, and ablation studies. Results consistently show Mambo’s superiority. Additional analyses (e.g., computational cost, OOD scoring strategies) add depth.

**Weaknesses:**

1. According to Table 1, compared to the baseline, the performance advantage of the method proposed in this paper is not significant. The method adds complexity (e.g., an extra prompt and hyperparameters).
2. Although Figure 5 visualizes the background prompt’s attention, a deeper and more analysis of the learned background concepts would strengthen interpretability.
3. According to [1], the similarity between the embeddings of background regions and the corresponding real label’s text embedding is greater than the similarity between the embeddings of foreground regions and the corresponding real label’s text embedding.  This proposed method does not take this into account.
4. The issue of inflexible background extraction strategies is mentioned in CLIP-OS [2].

[1] Li, Yi, et al. "A closer look at the explainability of Contrastive language-image pre-training." Pattern Recognition 162 (2025): 111409.
[2] Sun, Hao, et al. "Clip-driven outliers synthesis for few-shot OOD detection." arXiv preprint arXiv:2404.00323 (2024).

**Questions:**

When there are multiple foreground objects or the foreground is very small in an image, do Local Background Similarity and Local Similarity Refinement still remain effective?

---

> ### Author Response · Authors · 2025-11-20
>
> Thank you for taking the time to review our paper and for your constructive feedback. Below is our detailed response to your comments. All modifications will be reflected in the revised manuscript.
>
> >**Q1: According to Table 1, compared to the baseline, the performance advantage of the method proposed in this paper is not significant. The method adds complexity (e.g., an extra prompt and hyperparameters).**
>
> **A1:** **Regarding performance.** We acknowledge the observation in SCT that prompt-tuning based methods have nearly saturated improvements in AUROC, while there remains substantial room for improvement in FPR95; therefore, these two metrics should not be treated equally. Our improvements in FPR95 across various benchmark datasets are significant. Compared to SCT, our proposed method reduces FPR95 by **16.54%** on the ImageNet-100 benchmark, and decreases FPR95 by **12.05%** and **23.3%** under two different near-OOD task settings, respectively.
>
> **Regarding complexity.** (1) **Compared to OOD detection methods based on FG-BG decomposition**, our method introduces only **a small fraction of additional parameters** while achieving significant performance improvements. **Compared to negative prompt-based methods**, our method uses approximately **two hundredth** of the trainable parameters yet achieves **comparable performance**.  Moreover, we conducted additional experiments in Table 11 and the table below to verify the effectiveness of Mambo under the same number of parameters.
> | Method                 | Trainable Parameters (M) | FPR95 ↓        | AUROC ↑       |
> |------------------------|---------------------------|----------------|----------------|
> | LoCoOp                 | 0.0082                    | 29.01 ± 1.48   | 93.26 ± 0.27   |
> | SCT                    | 0.0082                    | 27.78 ± 1.56   | 92.82 ± 0.48   |
> | Local-Prompt           | 9.8300                    | 24.88 ± 0.24   | 94.40 ± 0.01   |
> | Mambo (N = 8, L = 8)   | 0.0082                    | 26.46 ± 0.48   | 93.51 ± 0.11   |
> | Mambo                  | 0.0410                    | 25.31 ± 0.30   | 93.68 ± 0.20   |
>
> From the results, it can be seen that even with **the same number of parameters**, our method still outperforms existing OOD detection methods based on FG-BG decomposition, which reflects the effectiveness and efficiency of our method. (2) In addition, we introduce a small number of hyperparameters to adjust different task objectives and better adapt to various downstream tasks. Experimental results show that Mambo achieves superior performance across different benchmarks. Therefore, introducing these hyperparameters is both important and necessary for improving the model’s generalization and robustness.
>
> >**Q2: Although Figure 5 visualizes the background prompt’s attention, a deeper and more analysis of the learned background concepts would strengthen interpretability.**
>
> **A2:** We thank the reviewer for the valuable feedback. While Figure 5 visualizes the attention of the background prompt, we **further** validate the learned background concepts through quantitative and more visualization. Specifically, we computed the cosine similarity between background text feature and class text features and used it to obtain the probability of the background feature for each class, **visualized in Figure 11**. The results show generally low and evenly distributed similarity, indicating that the learned background concepts are consistent and **class-agnostic across categories**.
>
> Additionally, we analyzed the **affinity of ID and OOD samples** to both class and background prompts. For each sample, the maximum similarity to class text features was taken as the class prompt affinity, and the similarity to background text feature as the background prompt affinity, **shown in Figure 12**. Results demonstrate that OOD samples generally have lower affinity to class prompts, while both ID and OOD samples show similar affinity to the background prompt. This confirms that the shared background prompt effectively captures class-agnostic background semantics without interfering with foreground classification boundaries. Slightly higher background affinity for OOD samples is likely due to their predominantly background content. These findings suggest that jointly leveraging class and background prompts enhances foreground semantics while mitigating background interference, leading to robust FS-OOD detection.

---

> ### Author Response · Authors · 2025-11-20
>
> >**Q3: According to [1], the similarity between the embeddings of background regions and the corresponding real label’s text embedding is greater than the similarity between the embeddings of foreground regions and the corresponding real label’s text embedding. This proposed method does not take this into account.**
>
> **A3:** We appreciate the reviewer for sharing an excellent work on in-depth analysis of foreground and background. **We will include this work in the related work section of the revised manuscript to provide a more comprehensive summary of prior studies.** **In fact, our method takes this into account and effectively addresses it.** In [1], the authors note that the high attention response of CLIP’s class attention map to background regions is mainly due to two factors:
>
> (1) the self-attention layers establish connections between semantically inconsistent regions, leading to spurious visualizations;
>
> (2) noisy activations are often associated with class-agnostic redundant features, and in that work, the authors mitigate noise by identifying features common across classes and suppressing their influence. **In fact, this idea is similar to FG-BG decomposition-based methods for OOD detection. These background features (i.e., OOD features) are also class-agnostic**, because unlike global features, CLIP’s local features are pre-pooling, meaning each region’s feature does **not mix concepts** and thus is not class-correlated due to label co-occurrence. Moreover, **these features are also redundant**; as noted in [2][3], classifiers relying solely on foreground features (i.e., ID-relevant features) can still achieve optimal decision boundaries. In our proposed Mambo, we introduce a background prompt shared across all classes, with one purpose being to identify class-agnostic background regions and then suppress their influence on classification using a maximum entropy loss, thereby mitigating noisy activations and avoiding spurious attention responses.
>
> >**Q4: The issue of inflexible background extraction strategies is mentioned in CLIP-OS.**
>
> **A4:** We thank the reviewer for sharing an excellent work that is highly relevant to FG-BG decomposition. We will include this work in the revised manuscript to provide a more comprehensive summary. In CLIP-OS [4], the authors indeed point out that LoCoOp’s background extraction strategy is **inflexible**. To address this issue, they propose the CLIP-surgery-discrepancy masking module, which cleverly leverages the property that foreground patches in the original CLIP have lower similarity to the true class, thereby enhancing the discrimination between foreground and background boundaries. However, CLIP-OS **differs from our work** in **several key aspects**. First, although the number of foreground patches can be flexibly adjusted based on similarity, **it still relies on a fixed threshold of 0.5 and cannot adaptively adjust according to the distribution of local features**. Second, **the motivation of our proposed method is to adaptively extract background regions according to different task objectives, which is not addressed in CLIP-OS through any module design**. This makes the motivation of our method **fundamentally different** from that of CLIP-OS.
>
> >**Q5: When there are multiple foreground objects or the foreground is very small in an image, do Local Background Similarity and Local Similarity Refinement still remain effective?**
>
> **A5:** **Yes**, when an image contains multiple foreground objects or when the foreground is very small, we observe that our FG-BG decomposition based on local background similarity remains effective. In the revised manuscript, **Figure 10** illustrates the final results of FG-BG decomposition under multiple foregrounds and small foreground conditions. The experimental results illustrate that even for these difficult cases, the local background similarity after the local similarity refinement **still focuses on background regions** while assigning low attention to different foreground objects. For example, in the tench class, where multiple fish overlap, the local background similarity is still able to correctly identify all foreground objects. In the nail class, although the snail occupies only a very small area, the local background similarity does not mistakenly classify additional background regions as foreground.
>
>
> [1] Li, Yi, et al. "A closer look at the explainability of Contrastive language-image pre-training." PR 162 (2025): 111409.
>
> [2] Ming, Yifei, Hang Yin, and Yixuan Li. "On the impact of spurious correlation for out-of-distribution detection." AAAI. Vol. 36. No. 9. 2022.
>
> [3] Miyai, Atsuyuki, et al. "Locoop: Few-shot out-of-distribution detection via prompt learning." NeurIPS 36 (2023): 76298-76310.
>
> [4] Sun, Hao, et al. "Clip-driven outliers synthesis for few-shot OOD detection." arXiv preprint arXiv:2404.00323 (2024).

---

> ### Author Response · Authors · 2025-11-27
> **Looking forward to your feedback**
>
> Dear reviewer uEb4:
>
> We respectfully appreciate again for your insightful and thoughtful comments! As the suggestions from the reviewer, we give thorough explanation and update the manuscripts accordingly.
>
> As it is approaching the end of the discussion period (December 3 at 11:59pm AoE) and we do not receive feedback, we sincerely hope you could look through our response and have a further comment at your convenience if you have any questions about the paper. We will do our best to address the issues of the reviewer.
>
> Best wishes,
>
> Submission 12004 Authors.

---

### Meta-Review · Area_Chair_wA7U · 2025-12-22

**Summary:**

The paper focuses on few-shot OOD detection and designs a background prompt to enhance the foreground–background discrimination under the few-shot setting. The manuscript received feedback from four reviewers: one reviewer posted acceptance but noted insufficiently strong supporting evidence, one reviewer gave a rejection due to issues with clarity and writing quality, and two provided prudent evaluations highlighting substantive concerns. The primary concerns raised by the latter two reviewers can be summarized as follows:
1. The proposed method appears to be an incremental refinement of the existing pipeline, lacking sufficiently distinctive or compelling innovation for the task.
2. The experimental results do not provide adequate evidence to support the claimed robustness of the method.
Following the rebuttal, none of the reviewers revised their scores. Although the authors presented additional illustrations, experiments, and discussions to address these concerns, the methodological innovation remains limited, and the experiments still fall short of demonstrating robustness in a convincing manner. Therefore, I recommend the rejection.

**Reviewer Concerns:**

Reviewer uEb4 shows concerns on complexity, visualization of background concepts, the similarity discussion, and inflexible background extraction, which are addressed by the rebuttal.

Reviewer VmDP requires a clear presentation, compelling performance, sufficient illustrations of motivation, and effectiveness under different backbones. The authors provide additional illustrations to support the motivation and effectiveness. However, I believe the paper actually requires a prudent refinement of methodological clarity and a compelling performance to show the robustness.

Reviewer vd43 requires adding the citation in Table 2, and the authors conduct the refinement.

Reviewer 7DqH posts throughout considerations on the innovations and the complexity. The authors address the complexity through performance comparison. The innovation problem is still outstanding.

**Reviewer Scores:**

After the rebuttal, none of the reviewers changed their scores. By considering the responses to the concerns raised by Reviewer 7DqH have not been sufficiently answered, I think the final ratings may be as follows:
uEb4 Rating: 5/ Confidence: 4
VmDP  Rating: 2 / Confidence: 5
vd43 Rating: 8 / Confidence: 3
7DqH Rating: 4 / Confidence: 4

---

### Decision · Program_Chairs · 2026-01-26

Reject